# Variability of cirrus cloud properties using a Polly$^{XT}$ Raman Lidar over high and tropical latitudes.

Kalliopi Artemis Voudouri[1], Elina Giannakaki[2,3], Mika Komppula[3], and Dimitris Balis[1]

[1]Laboratory of Atmospheric Physics, Physics Department, Aristotle University of Thessaloniki, Greece
[2]Department of Environmental Physics and Meteorology, Faculty of Physics, University of Athens, Greece
[3]Finnish Meteorological Institute, P.O.Box 1627, FI-70211, Kuopio, Finland

**Correspondence:** Kalliopi Artemis Voudouri (kavoudou@physics.auth.gr)

**Abstract.** Measurements of geometrical and optical properties of cirrus clouds, performed with a multi-wavelength Polly$^{XT}$ Raman Lidar, during the period 2008 to 2016 are analysed. The measurements were performed with the same instrument, during sequential periods, in three places at different latitudes, Gual Pahari (28.43°N, 77.15°E, 243m a.s.l) in India, Elandsfontein (26.25°S, 29.43°E, 1745m a.s.l) in South Africa and Kuopio (62.74°N, 27.54°E, 190m a.s.l) in Finland. The lidar dataset has been processed by an automatic cirrus cloud masking algorithm, developed in the frame of this work. In the following, we present a statistical analysis of the lidar retrieved geometrical characteristics (cloud boundaries, geometrical thickness) and optical properties of cirrus clouds (cloud optical depth, lidar ratio, ice crystal depolarization ratio) measured over the three areas that correspond to subtropical and subarctic regions as well as, their seasonal variability. The effect of multiple-scattering from ice particles to the derived optical products is also considered and corrected in this study. Our results show that cirrus layers, which have a noticeable monthly variability, were observed between 6.5 to 13km, with temperatures ranging from -72°C to -27°C. The observed differences on cirrus clouds geometrical and optical properties over the three regions are discussed in terms of latitudinal and temperature dependence. The latitudinal dependence of the geometrical properties is consistent with satellite observations, following the pattern observed with CLOUDSAT, with decreasing values towards the poles. Generally the geometrical boundaries have their highest values in the subtropical regions and overall, our results seem to demonstrate that subarctic cirrus clouds are colder, lower, and optical thinner than subtropical cirrus clouds. The dependence of cirrus clouds geometrical thickness and optical properties on mid temperature shows quite similar tendency for the three sites, but less variability for the subarctic dataset. Cirrus are geometrical and optical thicker at temperatures between -45°C and -35°C and a second peak is observed at lower temperatures $\sim$ -70°C for the subarctic site. Lidar ratio values also exhibit a pattern, showing higher values moving to the poles, with higher mean value observed over the subarctic site. The dependency of the mid temperature with the lidar ratio values and the particle depolarization values is further examined. Our study shows that the highest values of cirrus lidar ratio correspond to higher values of cirrus depolarization and warmer cirrus. The kind of information presented here can be rather useful in the cirrus parameterizations required as input to radiative transfer models, and can be a complementary tool to satellite products that cannot provide cloud vertical structure. In addition, ground-based statistics of the cirrus properties could be useful in the validation and improvement of the corresponding derived products from satellite retrievals.

## 1 Introduction

Cirrus clouds are usually formed in altitudes from 6 to 14km, having an average thickness of 1.5km and temperature variability from -80°C to -27°C (Westbrook et al., 2011). Cirrus are made predominantly, or entirely, of ice particles and the shape of their hydrometeors varies, affected by air temperature, atmospheric pressure, and ice supersaturation (Lynch et al., 2001). Given that cirrus clouds are challenging components in atmospheric and global climatological research, affecting the global radiation budget (e.g. Campbell et al., 2016), a number of studies have been focused on quantifying their geometrical, optical and microphysical properties (e.g. Seifert et al., 2007; Dionisi et al., 2013; Pandit et al., 2015). As systematic monitoring and accurate characterization of cirrus properties are important in the evaluation of models and satellite retrievals, a detailed monitoring of their properties at different geographical locations is crucial to understand their effects on climate.

Active remote sensing techniques, such as lidar and cloud radar instruments, have proved to be useful tools in providing continuous monitoring of high spatial and temporal distributions of cirrus clouds boundaries and their properties, and thus, enhancing the opportunity of tracking cloud evolution both in time and height. The capability of a cloud radar to map vertical and temporal structures of cloud layers has already been well recognized in the scientific community (Illingworth et al., 2007). Additionally, elastic backscatter and Raman lidars have also been used for retrieving geometrical and optical properties of cirrus clouds (i.e., Ansmann et al., 1992; Gouveia et al., 2017). Moreover, portable multiwavelength lidars (Polly$^{XT}$) allow for 24/7 monitoring of the atmospheric state (Engelmann et al., 2016) and can be used to establish long time series of aerosol or cloud measurements. Lidar observations also allow the retrieval of detailed hydrometeor properties, such as their sphericity, which is indicative of the shape of targets. The importance of ground-based lidar in monitoring cirrus clouds, is based on the mapping of particularly optically thin high altitude ice clouds which cannot produce sufficient reflectivity and as a consequence can be undetectable from cloud radars (Comstock et al., 2002) or from passive instruments. However, lidar beam attenuates strongly in liquid water clouds, and therefore, it is likely that in the case of multiple cloud layers reliable detection of cirrus clouds cannot be ensured.

In the last decades, observations of cirrus clouds properties have been conducted both in terms of field experiments (e.g. Seifert et al., 2007) and systematic observations (e.g. Dionisi et al., 2013; Pandit et al., 2015) from groundbased lidar systems, providing an estimation of their dependence on the geographical location. Dionisi et al. (2013) presented a methodology for identification and characterization of cirrus clouds properties, applied to the multiwavelength Rayleigh Mie-Raman (RMR) lidar in Rome. The study classified the detected cirrus clouds in different categories, based on their optical properties. Specifically, the analysis showed that 10% of the detected cirrus were subvisible clouds ($\tau < 0.03$), 49% thin ($0.03 < \tau < 0.3$) and 41% opaque cirrus ($\tau > 0.3$). The overall mean value of cirrus optical depth was calculated $0.37 \pm 0.18$, while the mean LR$_{eff}$ value was $31 \pm 15$ sr. Another statistical analysis on optical and geometrical properties of upper-tropospheric cirrus clouds based on a lidar dataset, was conducted in Amazonia (Gouveia et al., 2017). The frequency of occurrence of cirrus clouds classified as subvisible was 41.6%, whilst 37.8% was for thin cirrus and 20.5% for opaque cirrus. The correction of the multiple scattering

effect to the optical products in this study was made following the model of Hogan (2008). Lakkis et al. (2015) revealed that

the most commonly observed cirrus were characterized as optically thin cirrus, rather than opaque ones, with a mean optical depth value of 0.26 ± 0.11, over Buenos Aires (34.6°S, 58.5°W). There are also satellite based studies from either lidar (Cloud-Aerosol Lidar with Orthogonal Polarization (CALIOP), Dupont et al., 2010) or cloud radar (CloudSat) or combined lidar and cloud radar (e.g. Sassen et al. 2008) retrievals that provide a global view concerning the seasonal frequencies of cirrus clouds and their geometrical and optical properties and their variabilities.

However, there are only few long-term studies based on ground-based lidar systems, while these have a limited geographical distribution. This kind of observations that correspond to different areas and atmospheric conditions are crucial to reveal information of the latitudinal dependence of the cirrus properties and can provide indications about the aerosol effect on the geometrical and optical characteristics of the detected cirrus layers. On top of that, these observations can be further used in the validation and improvement of the satellite retrievals, which provide global distribution of cirrus clouds (Sassen et al., 2008).

Given that for satellite retrievals, the main input parameter to the optical processing of the cirrus layers is the lidar ratio, the selected lidar ratio value can introduce errors on the retrieved extinction and optical depth values of the cirrus layers, as it is illustrated by Young et al., (2018). The optical depth comparison of the Version 4.10 (V4) of the CALIOP optical depths and the optical depths reported by MODIS collection 6 show substantial improvements relative to earlier comparisons between CALIOP version 3 and MODIS collection 5, as a result of extensive upgrades of the extinction retrieval algorithm. New apriori

information of the lidar ratio value for the cirrus layers, included in Version 4.10 (V4) of the CALIOP data products, led to improvements of the extinction and optical depth estimates of the cirrus cloud layers. Thus, ground based lidar observations of the cirrus properties, that correspond to different areas and atmospheric conditions, are crucial to verify and eventually improve the satellite retrievals.

The aim of this work is to retrieve and analyze the cirrus geometrical, intensive and extensive optical properties at different

latitudes (subtropical and subarctic), from observations derived with the same ground based lidar system, which partly fills the gap concerning the latitudinal coverage of existing ground-based lidar studies. Then the observed differences are discussed in order to identify the possible causes. The information of the lidar ratio is an important parameter for the inversion of lidar signals in instruments that do not have Raman channel and space-based lidars, such as CALIPSO, and depend on a parameterization that may vary with location. Thus, information provided by well-calibrated ground based measurements is

quite critical. Analysis of the lidar ratios values derived from lidar measurements in different parts of the world, where different atmospheric and aerosol conditions prevail, will provide results that are more representative of the actual conditions and thus their use will lead to reductions in the uncertainties of the satellite retrievals.

The manuscript is structured as follows: after a brief description of the portable lidar system (Polly$^{XT}$) and the measuring sites in Section 2, we present the data analysis algorithm and the methods applied for the optical products retrievals in Section

3. The lidar derived statistical analysis and seasonal variations of geometrical and optical properties of cirrus clouds in both subtropical and subarctic areas over the period 2008-2016 are presented and discussed in Section 4. Concluding remarks are presented in Section 5.

## 2   Instrument and Measuring Sites

A multi-wavelength depolarization Raman lidar Polly$^{XT}$ of the Finnish Meteorological Institute (FMI) performed automated
measurements during the period 2008-2016 in three different geographical regions. The system is based on a compact, pulsed
Nd:YAG laser, emitting at 355, 532 and 1064 nm, at 20 Hz repetition rate. The laser beam is pointed into the atmosphere at an
off-zenith angle of 5°, so the impact of the specular reflection by ice crystals into cirrus layers on the backscattered signals is
negligible. The backscattered signal is collected by a Newtonian telescope, with 0.9m focal length. The vertical resolution of
the signal profiles is equal to 30m and the temporal resolution is 30s. The setup of the system includes two Raman channels
at 387 and 607 nm, three elastic channels at 355, 532 and 1064 nm, a depolarization channel at 355nm (for India and South
Africa), a water vapour channel at 407nm and depolarization channel at 532nm (cross-polarization with respect to the initial
emitted polarization plane) for Kuopio. Detailed description of the system is provided in Althausen et al., 2009 and Engelmann
et al., 2015. All measurements processed within the period 2008-2016 are available online at http://polly.tropos.de. A more
detailed description of the system components is presented in Table 1.

The Polly$^{XT}$ has participated in two campaigns in two subtropical areas, within the framework of the EUCAARI (European
Integrated project on Aerosol Cloud Climate and Air Quality interactions) project (Kulmala et al., 2011), covering a wide range
of cloud types. Measurements have been performed in Gual Pahari (28.43°N, 77.15°E, 243m a.s.l) in India from March 2008 to
March 2009 (Komppula et al., 2012), and in Elandsfontein (26.25°S, 29.43°E, 1745m a.s.l) about 150km from Johannesburg
in South Africa from December 2009 to January 2011 (Giannakaki et al., 2015). Figure 1 presents the map of the three
measuring sites. Measurements in Gual Pahari were not performed continuously from March 2008 to March 2009. Due to
technical problems with the laser, the data coverage from September to January was limited. Measurements could not be done
in October 2008 and January 2009, and in September and November-December only a few usable profiles were measured
(Komppula et al., 2012). Measurements in Elandsfontein were performed almost continuously, as two periods were dedicated
to the maintenance of the system (the one from December 23rd to January 26th 2009 and the second one, from October 23rd to
November 23rd 2010). Since November 2012 the Polly$^{XT}$ is operating in Kuopio (62.74° N, 27.54°E, 190m a.s.l) in Finland
providing continuous measurements and information for the presence of clouds of the subarctic area (Filioglou et al, 2017). The
three measurement sites constitute regions with different atmospheric conditions and different sources of aerosol and evaluation
of the cirrus dataset in these latitudinal experimental sites provides valuable information of the regional characteristics of the
measured cirrus properties.

# 3 Geometrical and optical retrievals of cirrus clouds

## 3.1 Description of the cirrus retrieval algorithm

Several steps were followed for the processing of the signal at 1064nm derived by the Polly$^{XT}$, needed for the estimation of the cirrus boundaries. These are illustrated in Figure 2. Firstly, the signal to noise ratio (SNR, Eq. 1) is calculated according to the following equation (Georgousis et al., 2015):

$$SNR = \frac{Csig}{\sqrt{Csig + Cbg}} \tag{1}$$

The SNR is selected above 3.5 (above this threshold value the boundary layers estimation found independent from SNR), since the lidar signal is strongly attenuated at higher altitude levels and the noisy parts of the signal should be rejected. Then, the zero and background levels are subtracted and the range-corrected signal is calculated. In the next step, we normalize the range-corrected signal by its maximum value found below 1.5km, so as to enhance the applicability of the method in various atmospheric conditions. Given that lidar signals are uncalibrated and signal levels from one lidar system to another can be rather different, the normalization ensures the applicability of the criteria used by Baars et al., 2008.

After these corrections are made, the Wavelet Covariance Transform (WCT) is applied to the range corrected signal. The method used (Eq. 2), detects discontinuities in the lidar signal, such as the top of the boundary layer, elevated aerosol layers or cloud boundaries, allowing the detection of cirrus cloud base and top (Brooks, 2003).

$$WCT = \sum_{b-\frac{\alpha}{2}}^{b} P(z)z^2 dz - \sum_{b}^{b+\frac{\alpha}{2}} P(z)z^2 dz \tag{2}$$

In Eq. 2, P(z) is the product profile where the WCT is being applied to. WCT is the result of the transformation, z is the altitude, b is the height at which a noticeable change in the normalized signal occurs, and $\alpha$ is the dilation chosen. A critical step to the accurate WCT application to the signal is the selection of an appropriate value of the window (dilation), so as to distinguish cloud layers from aerosol layers. In our case, a dilation of 225m, is chosen, proportional to the cirrus geometrical depth (Baars et al., 2008). Another critical step is the threshold WCT value for the determination of the cirrus boundary. A threshold value of 0.1 is selected as a detection limit for both the base (-0.1) and the top (+0.1) of cirrus cloud (Baars et al., 2008) after sensitivity studies. The WCT transformation has already been applied successfully on cirrus cloud detection (Dionisi et al., 2013).

Finally, cloud retrievals from the algorithm are classified as cirrus clouds when the following four criteria were met: i) the particle linear depolarization value is higher than 0.25 (Chen at al., 2002; Noel et al., 2002), ii) the altitude is higher than 6km and iii) the base temperature is below -27°C (Goldfarb et al., 2001; Westbrook et al., 2011) and iv) the top temperature is below -38°C (Campbell et al., 2015). The application of these criteria is made so as to avoid the presence of liquid water. It should be pointed out that lidar measurements were processed only in the absence of lower tropospheric (below 4 km) thick clouds.

The application of the WCT on a case of cirrus layer observed on July 20th 2016 at Kuopio station, for a time period between 00:00 and 01:00 UTC is presented. In our study, 60-min averages are computed and the respective mean value are taken as cloud base and top height. The hourly mean wavelet applied to the corrected 1064 signal and the hourly mean particle depolarization ratio and the backscatter coefficient profile of the cirrus evolution are presented in Fig. 3. The temperature values are also plotted with white line and the threshold values are marked with red lines.

### 3.2 Retrieval of the optical properties of cirrus

The integration of the extinction profile between the defined cloud base and the top of the cirrus layer is calculated to obtain the cirrus optical depth (COD) from the lidar measurements as shown in Eq. (3).

$$COD = \int\limits_{z_{base}}^{z_{top}} a_1(z)dz \tag{3}$$

The night-time measurements from Polly$^{XT}$ were processed by the Raman method, which allows the independent determination of the extinction and backscatter coefficients, thus providing the lidar ratio (extinction-to-backscatter ratio) (Ansmann et al., 1992). For the retrieval of the cirrus extinction coefficient profiles obtained from the daytime measurements, the integration of the backscatter profile multiplied by the lidar ratio is calculated. The daytime measurements from Polly$^{XT}$ were processed using the Klett inversion (Klett, 1981; Fernald, 1984), with respect to the ratio of the extinction to the backscatter coefficient. These two unknowns have to be related using either empirical or theoretical methods in order to be able to invert the lidar equation. In our study, the lidar ratio was determined by comparing the forward and the backward solution of Klett and the effective lidar ratio value was chosen as the value for which the aforementioned profiles tend to coincide (Ansmann et al., 1992). Both the daytime and night-time optical products were derived for each 1hour averaged profiles. The calculation of the corresponding molecular backscatter and extinction profiles was made based on temperature and pressure profiles obtained from radio soundings. Radiosondes launched daily at 06 and 18 UTC at the Jyvaskyla Airport, located to the southwest (62.39°N, 25.67°E) of the lidar station at Kuopio were used. Radiosonde observations released at Safdarjung Airport (28.58°N, 77.20°E) in New Delhi, India twice a day, and radiosondes from Upington International Airport (28.40°S, 21.25°E), in South Africa were used in the processing of the other two sites. Another important lidar quantity to be calculated is the particle depolarization ratio. This ratio constitutes a qualitative way to discriminate particle shapes and to distinguish spherical from non-spherical particles. Cirrus generally cause enhanced particle depolarization values, higher than 0.25 (see for e.g. Chen et al., 2002), depending on the ice-particle shape and orientation (Lynch et al., 2001). The calibration of the depolarization measurements, needed for the calculation of the particle depolarization ratio, was determined by using the geometric mean of the two ±45° measurements, following the procedure described by Freudenthaler et al., (2009). The particle depolarization ratio is presented only for the dataset of Kuopio, as for the other two sites only the Rayleigh calibration method for the calibration measurements was available.

## 3.3 Multiple scattering correction on optical products

The lidar equation assumes single scattering from the hydrometeor, but eventually the received photons could have been scattered multiple times before reaching the telescope. This effect, named multiple scattering, is considerably important primarily to the measured extinction coefficient values of cirrus clouds, and secondly to the calculated cirrus optical depth and the estimated lidar ratio values. Multiple scattering depends not only on cloud optical depth and cloud extinction, but also on the lidar system components, such as the laser beam divergence and the full-angle field-of-view of the receiver.

The relative influence of multiple scattering decreases with increasing height within the cloud, and the errors of the extinction coefficient can be even equal to 60% at the cirrus base (Lynch et al., 2001). As generally, multiple scattering effect cannot be negligible in a receiver field of view equal to 1mrad (Wandinger, 1998), this effect on cirrus clouds optical properties was considered and corrected in this study. In order to calculate the multiple scattering contribution to the calculated optical products, the Eloranta model (Eloranta, 1998) was used to estimate the ratio between the total received power and the contribution of

the single scattering, the ratio $P_{tot}(z)/P(z)$ (Eq. 4). The single extinction coefficient $a_1$ is then related to the actual (multiple scattering) coefficient a(z) through the parameter F as shown in Eq. (5) (Wandinger, 1998).

$$F_{(\lambda,z)} = \frac{\frac{d}{dz} ln \frac{P_{tot}(z)}{P(z)}}{2a_1(\lambda,z) + \frac{d}{dz} ln \frac{P_{tot}(z)}{P(z)}} \tag{4}$$

$$a_{(\lambda,z)} = \frac{a_1(\lambda,z)}{1 - F(\lambda,z)} \tag{5}$$

The model assumes cirrus consist of hexagonal ice crystals and the required inputs are: (i) the laser beam divergence, (ii)

the receiver field of view, (iii) the cirrus effective radius, (iv) the measured single scattering extinction profile (or the lidar ratio multiplied by the backscatter for the daytime measurements) and (v) the order of scattering. The estimation of the cirrus effective radius was taken from Wang and Sassen (2002), based on the linear relation of the effective radius with the cirrus cloud temperature derived from radio soundings. For the multiple scattering calculation, the code applies an iterative method including the following steps:

i) The measured extinction profile of the cirrus layer is provided ($a_1$).

ii) With the provided effective radius profile of the cirrus layer (linear relation of the effective radius with the cirrus temperature derived from radio soundings) and the measured extinction coefficient, an iterative procedure provides the ratio $P_{tot}$(z)/P(z).

iii) From (ii) a first value for the correcting factor F(z) can be worked out.

iv) The iterative procedure continues till the calculation of a stable correcting factor F(z) is found.

v) The corrected extinction can be then calculated from equation (5) and hence the value of lidar ratio.

The model has already been validated against other models (Hogan, 2006) in order to correct the derived optical characteristics of cirrus clouds and has already been applied in cirrus lidar applications (for e.g. Giannakaki et al. 2007). In the following

sections, the cirrus optical properties (lidar ratio, extinction coefficient, and optical depth) derived in the frame of this study were corrected for multiple scattering.

## 4 Results and discussion

In the following section, we present the mean geometrical and optical properties of the detected cirrus layers within the period 2008-2016 for the three measurement sites, which correspond to subtropical and subarctic regions, and we further discuss the differences between the retrieved properties.

### 4.1 Cirrus cloud cover detection

Cirrus cloud detection over the three regions is presented in Fig. 4. The detected cirrus clouds over Gual Pahari cannot provide any monthly trend and cannot be representative of an annual pattern. The time periods with technical issues (mentioned above) and the occurrence rate of low clouds observed between March and September leaded to a limited dataset of cirrus observations. Concerning the annual pattern observed over Elandsfontein, the maximum detection of cirrus layers is reported during May and December. No data processing could be performed during unfavourable weather conditions, such as the presence of low cloud, observed mainly the months between January to April with a percentage of $\sim 30\%$ of the total measurement period. The analysis of measurements over Kuopio showed that the cirrus cloud cover was found to vary both diurnally and seasonally. From the available data, the detection of cirrus clouds appears to exhibit an annual pattern with the maximum detection from February to September and minimum occurrence during the period between October and January, given the favorable meteorological conditions. Layers of low water clouds were present all year long, with the peak of monthly occurrence between April (28 cases) and November (27 cases). This monthly pattern of low clouds existence seems to follow the annual temperature cycle over the region (Jylha et al., 2004), with maximum temperature values observed during the period April to October. Concerning the diurnal pattern, the number of detected cirrus clouds during nighttime is higher from March to Semptember, and lower in the period from October to January.

### 4.2 Geometrical properties of cirrus clouds over the sub-arctic and tropical sites

Mean cirrus cloud geometrical thickness reported in literature from satellite retrievals is about 2.0 km globally (Sassen et al., 2008), while a broad distribution of geometrical boundaries from ground-based systems have been reported in literature (e.g., Gouveia et al., 2017; Seifert et al., 2007; Hoareau et al., 2013). Figure 5 shows the monthly variations of cirrus base height and the cirrus top height (displayed in monthly boxplots) derived with the automated algorithm, with the corresponding mean temperatures above each site. The cirrus geometrical properties show a broad monthly distribution ranging from 6790m to 13070, having the larger variability in the two subtropical sites compared to the subarctic site.

The cirrus lidar dataset in Gual Pahari (28.43°N, 77.15 °E,243m a.s.l - Northern hemisphere) region is the less extensive one compared to the other two sites and limitations due to the low signal to noise ratios exist. Indeed the sampling might not be statistically representative of the cirrus cloud properties, but some first results can be discussed. Specifically, during the

one-year-long measurement period, Polly$^{XT}$ was measuring on 183 days, corresponding to 2500h in total. The mean value of cirrus base is calculated 9000 ± 1580m, whilst mean top is found to be 10600 ± 1800m, with mean geometrical thickness of 1500 ± 700m. The temperature varied from -27°C to -50°C. Our results are consistent with another study over North China (Min et al., 2011), based on CALIOP satellite measurements. In this study a value of 1600 ± 1015m is reported for the cirrus geometrical thickness. According to this study, the cirrus top temperatures were found lower than -50°C and higher than -

80°C. A total measurement time of about 4935h corresponding to 88 cirrus profiles have been obtained over Elandsfontein (26.25°S, 29.43°E, 1745m a.s.l - Southern Hemisphere), during the observation period between 11 December 2009 and 31 January 2011, with the exception of the two periods of maintenance of the system (mentioned above). From the cirrus profiles processed, the mean value of cirrus base is calculated to be 9200 ± 810m, while mean top at 10826 ± 906m for the region of South Africa and the mean geometrical thickness is 1626 ± 735m. For the sub-arctic station of Kuopio (62.74°N, 27.54°E,

190m a.s.l), the seasonal mean cirrus cloud-base heights are calculated as follows: 8363 ± 1169m (MAM), 8326 ± 1120m (JJA), 9173 ± 1100m (SON), and 8900 ± 1390m (DJF) with an annual mean value of 8600 ± 1080m. The annual mean of the upper boundary of cirrus layers is 9800 ± 1075m, with a maximum value of 12595m during April. The mean geometrical thickness is calculated to be 1200 ± 700m. Base cirrus temperatures range from -71°C to -27°C having a mean value of -43°C. The corresponding temperature values of the top, range from -72°C to -38°C, with a mean value of -57°C. These values are

in accordance with the corresponding ones from the combined data of CloudSat and CALIPSO (Cloud-Aerosol Lidar and Infrared Pathfinder Satellite Observations) measurements (Sassen et al., 2008).

Table 2 summarizes the mean geometrical values calculated for each site separating daytime and nighttime measurements. The averaged geometrical properties between daytime and nighttime are found to be nearly identical above all sites, with differences less than 0.3km.

## 4.3 Optical properties of cirrus clouds over the sub-arctic and tropical sites

This section presents the cirrus optical properties for the three regions and Figure 6 shows the monthly variations of the cirrus optical properties (displayed in monthly boxplots) above each site.

The COD values over the three sites are presented in Figure 6a and 6b. For the subtropical region of Gual Pahari the mean COD 355 is 0.59 ± 0.25 and the mean COD 532 is found to be 0.45 ±0.30. The classification of clouds according to Sassen

and Cho (1992), shows that the detected cirrus layers are classified as follows: sub-visible cirrus (0%), optical thin cirrus (20%) and opaque cirrus (80%). One possible reason for the absence of subvisible cirrus clouds in this dataset can be the lower SNR that does not allow detectability of optically thin clouds at Gual Pahari. Another study over North China (Min et al. 2011), reported a mean value of optical depth of 0.41 ± 0.68 at 532nm and the classification of the detected cirrus layers was made as follows: subvisible cirrus (30.26%), optical thin (34.59%) and opaque cirrus (21.54%). Another study over the region (He

et al. 2013) reported that the optical depth of the cirrus layers varied between 0.0004 and 2.6, with a mean value of 0.33. For the subtropical region of Elandsfontein the mean value of COD 355 is calculated at 0.35 ± 0.03 and the mean COD 532 is found to be 0.30 ± 0.30. The COD have their highest values between April (1.36) and May (1.33) and December (1.02) and the percentage of 2% is categorized as subvisible cirrus, 61% as thin cirrus and 37% as opaque cirrus. For the Kuopio, the

column-integrated mean corrected COD at 355nm is $0.25 \pm 0.2$, and is found to vary between 0.018 and 1.53, while the mean

COD 532 is found to be $0.24 \pm 0.20$. The highest values of COD are found between January and March, with the highest value of 0.95. The mean COD 355 calculated in this study is larger than the value of $0.16 \pm 0.27$ reported by Das et al. (2009) and smaller than the value of $0.41 \pm 0.68$ reported by Min et al. (2011) from midlatitude observations. A number of other studies have reported mean COD values between 0.2 and 0.4. The mean cirrus optical depth reported for a tropical region is $0.25 \pm 0.46$ (Gouveia et al., 2017), for example, while the overall mean value of cirrus optical depth for a midlatitude station found

to be $0.37 \pm 0.18$ ( Dionisi et al., 2013). Reichardt (1998) reported that cirrus clouds optical depth values were lower than 0.3 for 70% of the cases processed for northern midlatitude cirrus. The classification of cirrus clouds according to Sassen and Cho (1992) indicates that 3% of the cirrus clouds measured in Kuopio are subvisible ($\tau < 0.03$), 71% are thin cirrus ($0.03 < \tau < 0.3$) and 26% are opaque cirrus ($\tau > 0.3$). The low percentage of the subvisible category of cirrus layers, have also been observed over midlatitude sites (e.g., Kienast-Sjogren et al., 2016), where subvisible cirrus clouds have been measured during 6% of the

observation time.

In what follows we proceed into finding a connection between the COD values derived in the different sites and the AOD load over the regions which are exposed to different aerosol sources. Table 4 lists the predominant aerosol type over each region and the results from the analysis of AOD at 355 nm in the free troposphere and the calculated COD values. We can conclude that there is an indication of the relationship of the aerosol load on the derived cirrus statistics, as the higher AOD values are

linked with the higher COD values calculated for the two subtropic regions. More specifically, the one year aerosol analysis of lidar observations in Gual Pahari (Komppula et al., 2012) showed that in the summer, the measured air masses were slightly more polluted and the particles were a bit larger than in other seasons (higher Angstrom exponent values), with the main aerosol sources to be the local and regional biomass and fossil fuel burning. The annual averages revealed a distinct seasonal pattern of aerosol profiles, with aerosol concentrations slightly higher in summer (June - August) compared to other seasons, and

particles larger in size. During the summer and autumn, the average lidar ratios were larger than 50 sr, suggesting the presence of absorbing aerosols from biomass burning. The lidar observations that were performed at Elandsfontein and used for aerosol characterization for the corresponding study period (Giannakaki et al. 2016) showed that the observed layers were classified as urban / industrial, biomass burning, and mixed aerosols using the information of backward trajectories, MODIS hotspot fire products and in situ aerosol observations. The analysis of the seasonal pattern of vertical profiles of the aerosol optical

properties showed that the more absorbing (higher lidar ratio at 355 nm) biomass particles were larger on August and October, while the category of Urban/industrial had their peak on January, March and May. Kuopio is an urban area and constitutes a low aerosol content environment. The columnar analysis of sunphotometer observations (Aaltonen et al., 2010) revealed that the high Angstrom exponent values observed can be possible linked with the presence of fine particles, while the seasonal analysis of the optical depth showed that there is no significant variation.

Concerning the lidar ratios values (Figure 6c and 6d) observed over Gual Pahari, the lidar ratio value at 355nm is calculated at $27 \pm 12$ sr and the corresponding one for 532nm is $28 \pm 22$ sr and the lidar ratios reach their highest values on May. Our results are in agreement with another cirrus cloud study for the area; He et al. (2013) report a mean lidar ratio value of 28 sr, using a micropulse lidar. For the Elandsfontein site, the mean LR 355 value is found to be $26 \pm 6$ sr and the mean LR

532 is 25 ± 6 sr and the lidar ratios reach their highest values during April. A mean value lidar ratio of 33 ± 7 sr at 355nm is observed for the whole period studied over Kuopio site, with higher variability observed on June, while the corresponding mean value LR 532 is calculated to be 31 ± 7sr, without any obvious seasonal cycle. Specifically, the mean LR 355 for the corresponding months are calculated as follows: 33 ± 7 sr (MAA), 34 ± 7 (JJA), 33 ± 7 (SON) and 34 ± 5 (DJF). For opaque, thin and sub-visible cirrus clouds the means are 31 ± 6 sr, 34 ± 7 sr, and 35 ± 7 sr, respectively. Gouveia et al. (2017) found a mean LR 355 value of 23.9 ± 8.0sr (SD) for the tropical region of Amazonia, while Giannakaki et al., (2007), reported a corresponding value of 30 ± 17sr for a mid-latitude station. Josset et al. (2012) and Garnier et al. (2015) analyzed spaceborne CALIOP (Cloud Aerosol Lidar with Orthogonal Polarization) lidar observations. Both studies concluded that cirrus lidar ratio (corrected for multiple scattering effects) around the globe has typically values of 30-35sr ± 5-8sr at 532 nm. Nevertheless, the lidar ratio values may vary greatly depending not only on the altitude and composition of the cirrus clouds (Goldfarb et al., 2001), but also on the correction of the multiple scattering effect (Platt, 1981; Hogan, 2008). The aforementioned depends on the ice crystals effective radius and the associated uncertainty could range from 20 to 60% (Wandinger, 1998). Lidar ratio for cirrus clouds was assumed to be constant with altitude and season with a value of 25 sr using the CALIOP extinction retrieval algorithm (Young et al., 2013; Young and Vaughan, 2009), but this value has changed in the upgraded algorithm, as illustrated by Young et al. (2018).

Concerning the monthly variability of the depolarization values (Figure 6e) over Kuopio, no clear tendency is observed. The higher monthly mean value was observed on July, but the variability was less than 0.04 between months, with a mean value of 0.38 ± 0.07.

As the assumption that the backscatter and the extinction coefficients for sufficiently large cirrus particles are spectrally independent; the color ratio (ratio of backscatter profiles, CR) at 355 and 532 is supposed to be equal one. This assumption is also used in satellite processing schemes. However, it is reported that the measured variability of cirrus color ratios is much larger than previously realized and that measured color ratios are higher in the tropics (Vaughan et al., 2010). For the Kuopio station, mean CR is found 1.1 ± 0.8, while for the less extensive dataset of New Delhi the mean value is found 1.5 ± 0.8 and for Elandsfontein the mean value is 1.4 ± 1.1.

Table 3 summarizes the mean optical values discussed above, for the three sites, separating daytime and nighttime observations. Generally, the averaged optical properties values are found to be nearly identical, except one site (New Delhi), where average nighttime optical properties found higher (∼ 4sr) than that of daytime.

To further investigate the distribution of the cirrus lidar ratio values over Kuopio, we present a histogram of the values derived in Fig. 7. The most frequent observed lidar ratio values ranging between 28 and 36 sr for 355 nm and 20 and 36 sr for 532 nm. Similar results have been retrieved regarding the variability of LR 532, which is constant from one month to another, as shown. This figure can provide an evidence that although the lidar dataset are not continuous (due to not favour weather conditions the winter months), the frequency distributions are close to normal and thus the statistics shown here have a significance. In addition we can claim that with this scarce sample of data we observe consistent results with a number of other literature studies.

In Figure 8, we examine the dependence of the LR 355 with the COD 355 values on intervals of 5 sr. The dashed lines indicate the categories defined by Sassen and Cho (1992). The most common lidar ratio values from 25 to 40 sr are found for quite low COD values (corresponding to thin cirrus) for the subarctic station.

### 4.4 Cirrus classification at Kuopio

The classification of cirrus clouds according to Sassen and Cho (1992) is made based on the COD values. Ground-based lidars are well suited for thin cirrus layers observations, due to their sensitivity to thin atmospheric features, in contrast to spaceborne lidar observations (Martins et., 2011). For this reason, additional analysis on each cirrus category is also conducted for Kuopio site as measurements in this station represent the most extensive dataset between November 2012 and December 2016.

#### Category "Subvisible"

Subvisible cirrus are geometrical thin layers with mean geometrical thickness of $643 \pm 211$ m. Generally, subvisible cirrus detection is a challenging component in satellite retrievals. MODIS, for example, is not sensitive to optically thin cirrus clouds due to the insufficient contrast with the surface radiance (Ackerman et al., 2008; Ackerman et al., 2010), while the CALIPSO and CloudSat observations are more sensitive to the height and presence of subvisible and thin cirrus (Hong et al., 2010). Thus, the mapping of subvisible cirrus can be rather important in climatological studies. In our study, 6 cases of cirrus with COD less than 0.03 are analyzed, mostly detected during February. Subvisible cirrus geometrical thickness found $750 \pm 269$ m, less than the mean value of all cirrus clouds, and their temperature $2°C$-$3°C$ colder than the mean temperature. These values are consistent with previous studies of subvisible cirrus from spaceborne lidar observations, examined on a global scale (Martins et., 2011). Their mean COD 355 is calculated $0.021 \pm 0.0031$, their mean LR 355 is $34 \pm 7$sr and their mean particle depolarization value is 0.45.

#### Category "Thin"

As mentioned previously, thin cirrus is the most predominant type of cirrus in our study, with 152 observations. Thin cirrus can also be undetectable by passive remote-sensing satellites, especially the ones with COD less than 0.2, and have so far not systematically been characterized. Their geometrical thickness found to be $1100 \pm 586$ m. Their mean COD 355 is calculated $0.16 \pm 0.07$, their mean LR 355 is $34 \pm 7$ sr and their mean particle depolarization value is $0.3 \pm 0.13$.

#### Category "Opaque"

Opaque cirrus are the one with the highest value of optical depth that contribute the most of the total radiative forcing (Kienast-Sjogren et al., 2016). In our study, a total of 55 measurements of opaque cirrus are processed. Their mean geometrical thickness is found to be $1462 \pm 659$ m, higher than the value of all cirrus categories. Their mean COD 355 is calculated $0.5 \pm 0.21$, their mean LR 355 is $31 \pm 6$ sr and their mean particle depolarization value is $0.33 \pm 0.12$.

## 4.5 Latitudinal and temperature dependence of cirrus properties

The three presented datasets are derived from different latitudinal and climatic sites. In this section we firstly examine the latitudinal dependence of the cirrus geometrical and optical properties. The reported values in literature from previous studies based on lidar grounbased dataset and the retrievals of the current one are listed in Table 5 and plotted in Figure 9 for comparison. We can note, that the cirrus geometrical properties and the lidar ratio values may vary greatly depending on the latitude and an decreasing trend of the geometrical boundaries with the rise of the distance from the equator is obvious, also reported by satellite observations (Sassen et al., 2008). Generally, cirrus layers have been observed up to altitudes of 13km above the subtropical sites, whereas they have only been detected to about 1km lower at the subarctic region and this conclusion is in accordance with the Cloudsat observations (Sassen et al., 2008). Based on the satellite information, the derived cirrus cloud thicknesses was found to be larger in the tropics and decreasing toward the poles. Also from the values reported from groundbased studies, a pattern can be concluded: cirrus cloud geometrical properties peaks around the equator and at midlatitudes sites, with generally decreasing amounts as the poles are approached. On the other hand, the lidar ratio values seem to follow a diverse relation, showing greater values moving to the poles. In our study, lower LR are observed for Gual Pahari and Elandsfontein and higher mean value for Kuopio. The larger variability of the optical properties at the two subtropical regions, relative to Kuopio, could be related to the larger and variable aerosol load over these regions. Overall, our results seem to demonstrate that subarctic cirrus clouds are colder, lower and optical thinner than subtropical cirrus clouds. However, a more extended database is needed to strengthen these indications.

The dependence of geometrical and optical properties on mid-cirrus temperature is also examined in Fig. 10. In order to investigate this dependence, we have grouped cirrus clouds temperatures into 5°C intervals. The number of cases per temperature bin are also labeled. Temperature values are obtained from radio soundings, as mentioned above. Thicker clouds ( $\sim$ 1.5 km) are observed at temperatures between $\sim$ -45°C and $\sim$ -35°C, with decreasing thickness reported for lower temperatures, for both the subtropic and subarctic regions and a second peak is found in the range between $\sim$ -75°C and $\sim$ -65°C for the subarctic station. A similar trend has been reported for a midlatitude region by Hoareau et al. (2013), where thickest cirrus layers were found about -42.5°C, and thinner ones at both colder and warmer temperatures. Another study (Pandit et al., 2015) reports that the geometrical thickness increases from 1 to 3.5 km as mid-cloud temperature increases from -90 to -60°C, while for the further increase in temperature from -60 to -20 °C, the geometrical thickness decreases to less than 1 km. Concerning the optical properties shown in Fig. 10, a steady increase of lidar ratio from -25°C to -40°C is noticed for the two subtropical stations,while the variability of this parameter is relatively constant across months for the subtropic station, with a slightly increase at warmer temperatures (Figure 10b). There are indications that the cloud optical depth increases with the increasing cirrus mid temperature for the two subtropical sites (Fig. 10c). At cold temperatures ( $\sim$ -65°C), optical thickness for cirrus layers of the subarctic station is high, compared to warmer temperatures and also cloud thickness for this temperature is similar high ( $\sim$ 1.5 km). The dependence of the particle depolarization values on base temperature is also examined (Fig. 10d). No clear tendency is found, as the variability of this parameter is relatively constant, with a slightly increase of the particle

depolarization with the increasing mid temperature. This behaviour indicates a relation between cirrus ice crystal shape and temperature, however, more studies should be done in order to examine this behaviour on various geographical locations.

Figure 11 presents the color ratios values on 5°C intervals of cirrus mid temperature, indicating an almost stable behavior with temperature. Generally, we can conclude that for higher altitudes, lower spectral dependence is noticed, taking also into account the number of measurements performed at each site.

The dependency of the mid temperature with the lidar ratio values at 355nm and the particle depolarization values is further examined (Figure 12). Fig. 12 shows that the highest values of cirrus lidar ratio (>40) correspond to higher values of cirrus

depolarization (>0.4) and warmer cirrus. Moreover, it can be seen the variety of depol values that correspond to the mean value of lidar ratio ($\sim$ 31). A similar behavior is reported in Chen et al. (2002) for lidar ratio values higher than 30 sr. In his study, the relationship between the depolarization ratio and the lidar ratios shows the former split into two groups for lidar ratios higher than 30. The first group has high depolarization ratios about 0.5 and the second one has 0.2.

## 5   Conclusions

Observations of cirrus clouds geometrical and optical properties, performed with a ground-based multi-wavelength Polly$^{XT}$ Raman Lidar, during the period 2008 to 2016 are analyzed and presented in this study. The measurements were performed in three places at different latitudes, Gual Pahari (28.43°N, 77.15°E, 243m a.s.l) in India, Elandsfontein (26.25°S, 29.43°E, 1745m a.s.l) in South Africa and Kuopio (62.74 °N, 27.54°E, 190m a.s.l) in Finland and an algorithm is developed to automatically define the cirrus clouds boundaries.

The statistical behaviour of the cirrus clouds properties in the different geographical and climatic counterparts shows that the geometrical boundaries display large distribution for the two subtropical regions with higher values of geometrical thickness, with mean thickness of $1500 \pm 700$m, $1600 \pm 735$m and $1200 \pm 700$m for Gual Pahari, Elandsfontein and Kuopio respectively, showing their dependence on the geographical location. The corresponding overall mean value of COD 355 is calculated to be $0.60 \pm 0.25$ and $0.35 \pm 0.30$, for Gual Pahari and Elandsfontein correspondingly, while a slightly lower mean of $0.25 \pm 0.2$

is calculated for Kuopio. The lidar ratio values at 355nm show higher values moving to the poles, with calculated values to be $27 \pm 12$ sr, $26 \pm 6$ sr, and $33 \pm 7$ sr for Gual Pahari, Elandsfontein and Kuopio, respectively. Overall, our results seem to demonstrate that subarctic cirrus clouds are colder, lower, and optical thinner than subtropical cirrus clouds. However, a more extended database is needed to strengthen these indications.

The dependence of cirrus clouds geometrical thickness and optical properties on mean temperature is also examined, showing

quite similar tendency, but less variability for the subarctic dataset. The dependence of cirrus clouds geometrical thickness and optical properties on mid temperature shows quite similar tendency, but less variability for the subarctic dataset. Cirrus found geometrical and optical thickest at temperatures between -45°C and -35°C. At temperatures below -55°C, the optical thickness of cirrus layers becomes again high and this trend appears only for the subarctic station. However, we should keep in mind that the number of samples corresponding to temperatures below -60°C is limited. The lidar ratio is found to be quite constant

with temperature, with a slightly increase in the warmer mid temperatures, showing larger variability for the subtropic datasets, while the particle depolarization values seem almost constant at temperatures between -27°C and -60°C.

The geometrical and optical properties of cirrus layers are studied in detail, providing information useful in the validation of the cirrus parameterizations in models. Furthermore, our results could be useful for lidar ratio selection schemes needed by satellite optical properties retrievals of cirrus layers over different locations, e.g., the upcoming EarthCARE (Earth Cloud Aerosol and Radiation Explorer) mission. The spectral dependence discussed above, is another important issue for the satellite algorithm schemes, given the different wavelengths applied among the different satellites.

In any case, further cirrus observations must be conducted, so as to investigate whether differences in the background aerosol load contribute to potential differences in the cirrus cloud geometrical and optical properties and which are the different atmospheric mechanisms leading to these differences over the different regions.

*Author contributions.* KA. Voudouri prepared the automatic algorithm for the cirrus detection and processed the lidar measurments for the optical retrievals during the period 2008-2016. KA. Voudouri prepared the figures of the manuscript. E. Giannakaki reviewed parts of the results. M.Komppula is the PI of the lidar station and D. Balis directed the preparation of the manuscript. KA. Voudouri prepared the manuscript with contributions from all co-authors.

*Competing interests.* The authors declare that they have no conflict of interest.

*Acknowledgements.* This work has been conducted in the framework of EARLINET (EVR1 CT1999-40003), EARLINET ASOS (RICA-025991), ACTRIS, and ACTRIS-2 funded by the European Commission. The research leading to these results has received funding from the European Union's Horizon 2020 research and innovation program under grant agreement no. 654109 and previously from the European Union Seventh Framework Programme (FP7/2007-2013) under grant agreement no. 262254. This work was partly funded by the European Commission 6th Framework under the European Integrated Project on Aerosol Cloud Climate and Air Quality Interactions, EUCAARI. Voudouri K.A acknowledges the support of the General Secretariat for Research and Technology (GSRT) and Hellenic Foundation for Research and Innovation (HFRI). (Scholarship Code: 95041).

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

**Table 1.** Technical specifications of the Polly$^{XT}$ System.

| Characteristics | Polly$^{XT}$ |
| --- | --- |
| Operating Wavelength(s) | 355nm, 532nm, 1064nm |
| Average pulse energy | ~450 mJ |
| Laser beam divergence | <0.2 mrad |
| Telescope diameter | 0.3m |
| Receiver field of view | 1mrad |
| Detectors | Hamamatsu PMTs |
| Polarization | Cross & Total |
| Raw data range resolution | 30m |
| Raw data time resolution | 30s |

**Table 2.** Average cirrus properties for the three regions for daytime and nighttime measurements.

| Cirrus Properties | Gual Pahari | Elandsfontein | Kuopio |
|---|---|---|---|
| Cirrus Base (m) | $9000 \pm 1580$ | $9200 \pm 810$ | $8600 \pm 1080$ |
| | $8900 \pm 1480$ d | $9200 \pm 818$ d | $8037 \pm 914$ d |
| | $9000 \pm 1529$ n | $9200 \pm 744$ n | $7900 \pm 1246$ n |
| Cirrus Top (m) | $10600 \pm 1800$ | $10826 \pm 906$ | $9800 \pm 1075$ |
| | $10350 \pm 2000$ d | $10705 \pm 928$ d | $9443 \pm 1095$ d |
| | $10900 \pm 1700$ n | $10889 \pm 928$ n | $8965 \pm 1055$ n |
| Cirrus geometrical thickness (m) | $1500 \pm 700$ | $1600 \pm 735$ | $1200 \pm 700$ |
| | $1480 \pm 700$ d | $1627 \pm 802$ d | $1167 \pm 700$ d |
| | $1300 \pm 638$ n | $1696 \pm 616$ n | $1243 \pm 700$ n |
| Temperature base (C) | $-33 \pm 6$ | $-34 \pm 5$ | $-43 \pm 10$ |
| Temperature top (C) | $-45 \pm 4$ | $-45 \pm 6$ | $-57 \pm 9$ |

**Table 3.** Average cirrus optical properties for the three regions for daytime and nighttime measurements.

| Cirrus Properties | Gual Pahari | Elandsfontein | Kuopio |
| --- | --- | --- | --- |
| N | 11 (7d, 4n) | 64 (32d, 32n) | 213 (153d, 50n) |
| % subvisible | 0 | 2 | 3 |
| % thin | 20 | 61 | 71 |
| % opaque | 80 | 37 | 26 |
| LR 355 | $27 \pm 12$ | $26 \pm 6$ | $33 \pm 7$ |
|  | $23 \pm 8$ d | $24 \pm 7$ d | $33 \pm 7$ d |
|  | $31 \pm 15$ n | $27 \pm 8$ n | $33 \pm 7$ n |
| LR 532 | $28 \pm 22$ | $25 \pm 6$ | $31 \pm 7$ |
|  | $23 \pm 3$ d | $24 \pm 5$ d | $31 \pm 7$ d |
|  | $33 \pm 11$ n | $26 \pm 7$ n | $30 \pm 7$ n |
| COD 355 | $0.60 \pm 0.25$ | $0.35 \pm 0.30$ | $0.25 \pm 0.20$ |
|  | $0.40 \pm 0.30$ d | $0.34 \pm 0.30$ d | $0.24 \pm 0.21$ d |
|  | $0.80 \pm 0.20$ n | $0.36 \pm 0.30$ n | $0.26 \pm 0.20$ n |
| COD 532 | $0.45 \pm 0.30$ | $0.30 \pm 0.30$ | $0.24 \pm 0.20$ |
|  | $0.30 \pm 0.40$ d | $0.25 \pm 0.30$ d | $0.26 \pm 0.20$ d |
|  | $0.60 \pm 0.20$ n | $0.35 \pm 0.30$ n | $0.22 \pm 0.20$ n |
| CR (355/532) | $1.50 \pm 0.80$ | $1.40 \pm 1.10$ | $1.10 \pm 0.80$ |

**Table 4.** Predominant aerosol type and AOD FT for the three regions.

| Measurement Site | Gual Pahari | Elandsfontein | Kuopio |
|---|---|---|---|
| Predominant aerosol type | dust particles , biomass burning | biomass burning, desert dust particles and urban particles | fine particles |
| AOD FT | $0.09 \pm 0.03$ | $0.06 \pm 0.04$ | $0.01 \pm 0.01$ |
| COD | $0.60 \pm 0.25$ | $0.35 \pm 0.30$ | $0.25 \pm 0.20$ |

**Table 5.** Summary of cirrus clouds geometrical and optical properties of ground-based lidar observations reported in literature.

| Measurement site | Location | Cirrus Base (km) | Cirrus Top (km) | LR (sr) | COD | Reference |
|---|---|---|---|---|---|---|
| Kuopio | 62.74°N, 27.54°E | 8.0 ± 1.1 | 9.3 ± 1.1 | 33 ± 7 | 0.25 ± 0.20 | This Study |
| France | 43.9°N, 5.7°E | 9.3 ± 1.8 | 10.9 ± 1.7 | | | Hoareau et al., 2013 |
| Rome | 41.8°N, 12.6°E | | | 31 ± 15 | 0.37 ± 0.18 (532nm) | Dionisi et al., 2013 |
| Thessaloniki | 40.6°N, 22.9°E | 8.8 ± 1.0 | 10.3 ± 0.9 | 30 ± 17 | 0.31 ± 0.24 (355nm) | Giannakaki et al., 2007 |
| Naqu | 31.5°N, 92.1°E | 13.7 ± 2 | 15.6 ± 1.6 | 28 ± 15 | 0.33 ± 0.29 (532nm) | He et al., 2013 |
| Gual Pahari | 28.43°N, 77.15°E | 9.0 ± 1.6 | 10.6 ± 1.8 | 27±12 | 0.59 ± 0.39(355nm) | This Study |
| Gadanki | 13.5°N, 79.2°E | 13.0 ± 2.2 | 15.3 ± 2.0 | | | Pandit et al., 2015 |
| Hulule | 4.1°N, 73.3°E | 12 ± 1.6 | 13.7 ± 1.4 | 32 | 0.28 (532nm) | Seifert et al., 2007 |
| Amazonia | 2.89°S, 59.97°W | 12.9 ± 2 | 14.3 ± 1.9 | 23 ± 8 | 0.25 ± 0.46 (355nm) | Gouveia et al., 2017 |
| La Reunion | 20.8°S, 55.5°W | | 13.0 | | 0.05 | Hoareau et al., 2012 |
| Elandsfontein | 26.25°S, 29.43°E | 9.2 ± 0.8 | 11 ± 0.9 | 26 ± 6 | 0.35 ± 0.30 | This Study |
| Buenos Aires | 34.6°S, 58.5°W | 9.6 | 11.8 | | 0.26 ± 0.11 | Lakkis et al., 2015 |

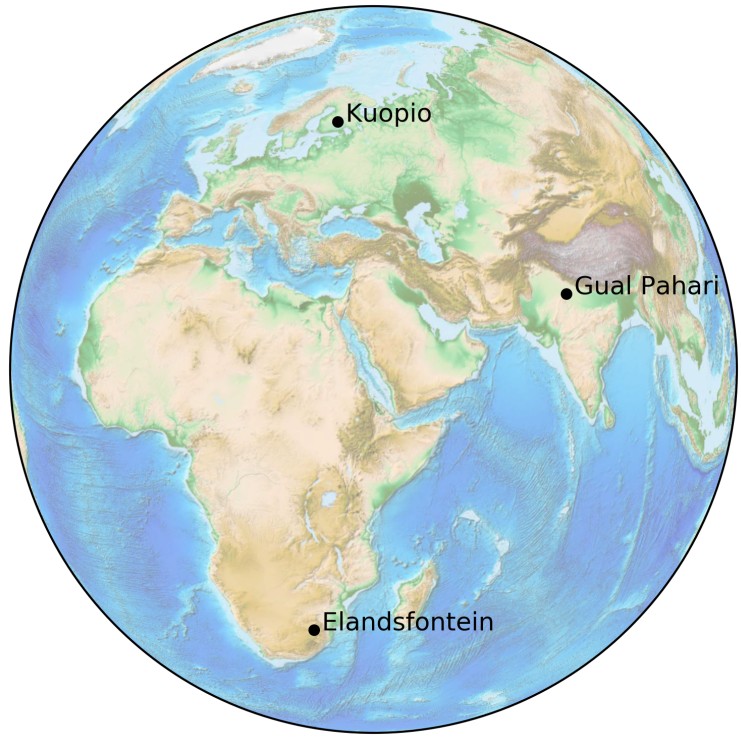

**Figure 1.** Map of the three measuring sites.

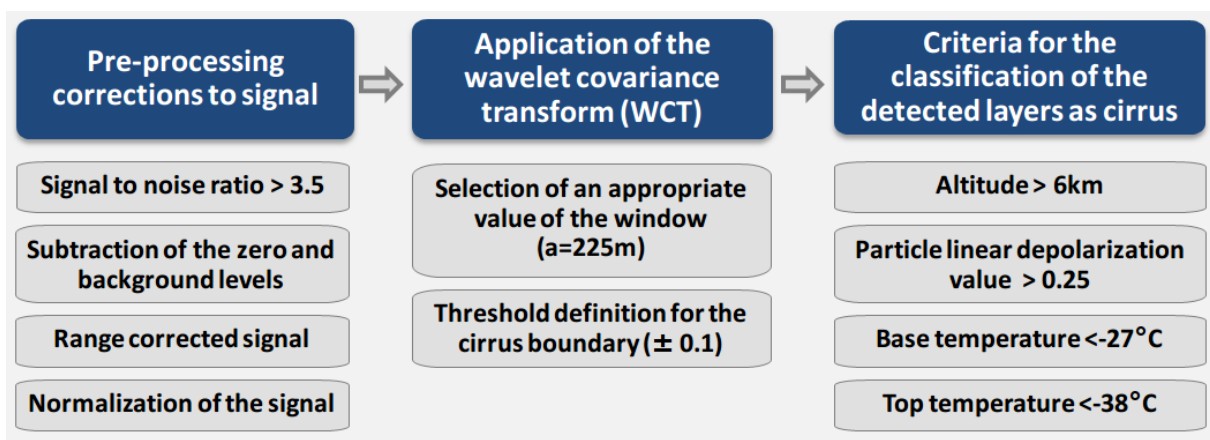

**Figure 2.** Schematic flowchart showing the main steps of the methodology applied in this study to obtain the cirrus geometrical boundaries from the Polly$^{XT}$ measurements.

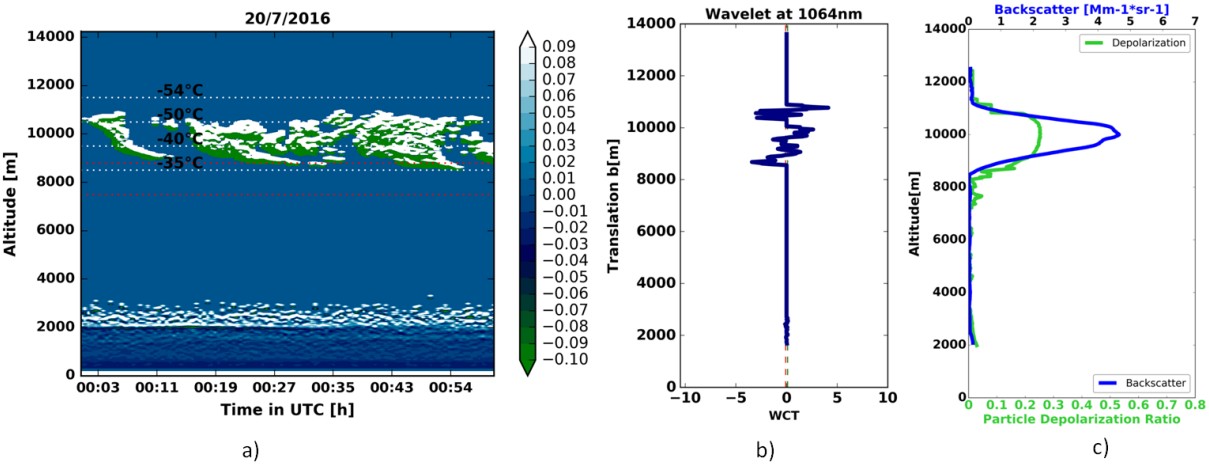

**Figure 3.** Cirrus cloud evolution as determined from the Polly$^{XT}$ for 1 hour observation with temperatures marked with white lines and the temperatures criteria marked with red lines (a), the 1-hour averaged wavelet applied to the corrected 1064 signal (b) and the hourly mean particle depolarization ratio (green) and backscatter coefficient (blue) at 532nm (c) on 20th of July 2016 at Kuopio station.

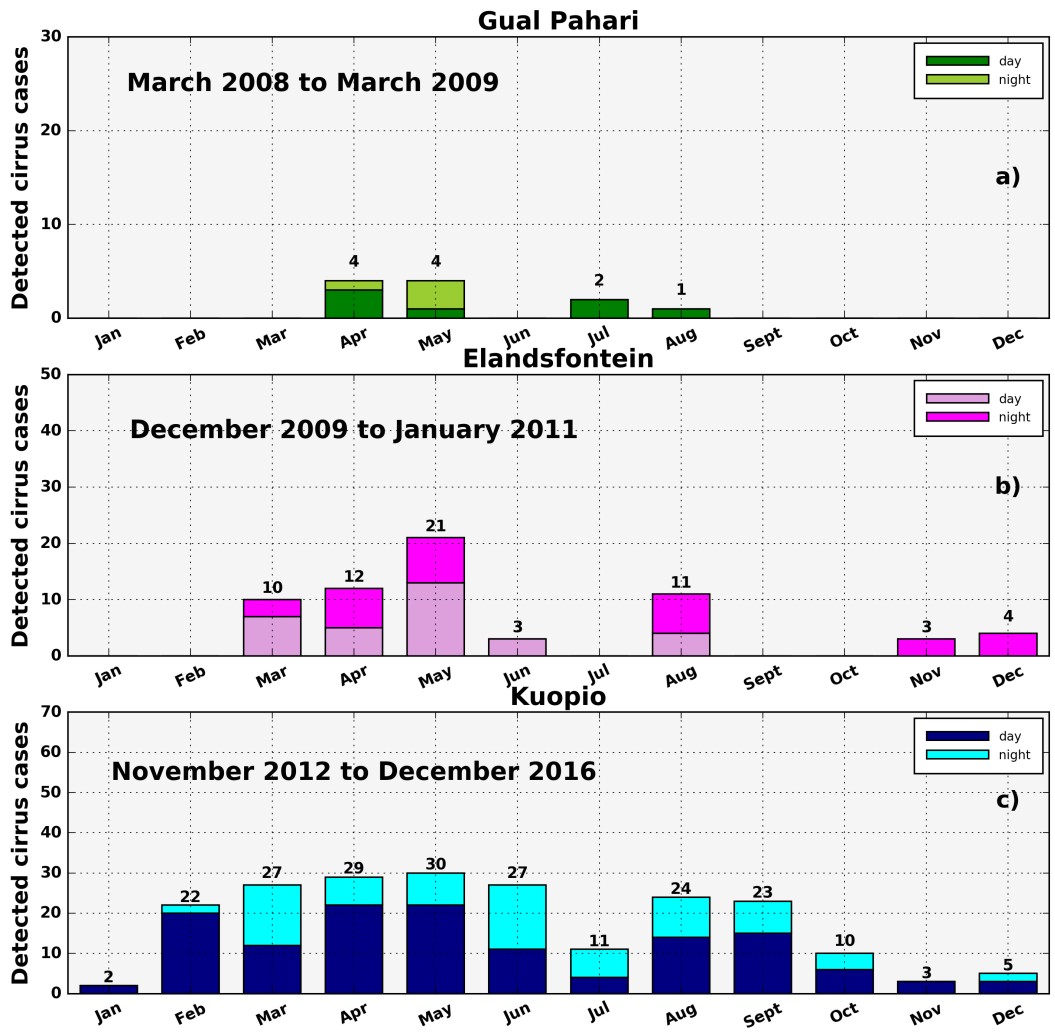

**Figure 4.** Cirrus detection with the Polly$^{XT}$ over Gual Pahari, India during the period 2008-2009 (a) Elandsfontein, South Africa during the period 2009-2010(b) and Kuopio (c), Finland during 2014-2016.

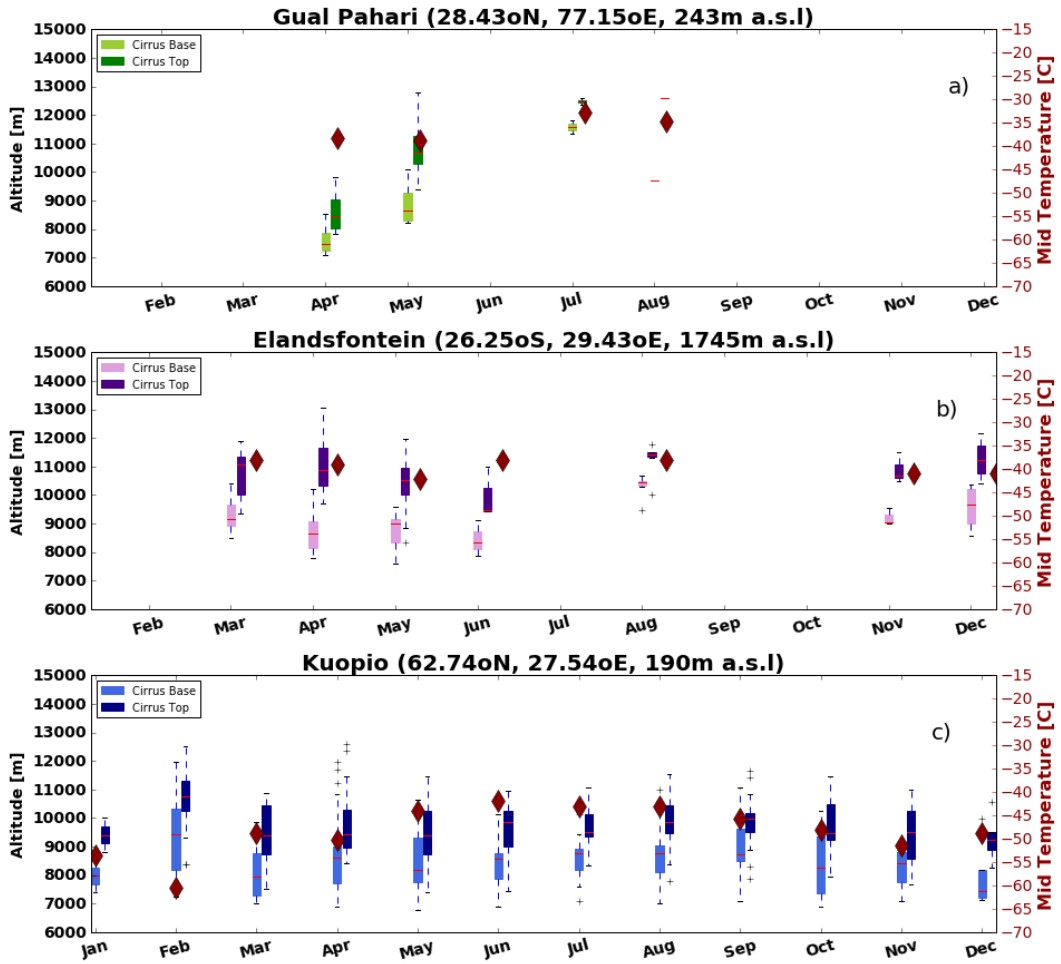

**Figure 5.** Monthly cycle of mean base, mean top and the corresponding temperature base values (circles) of the cirrus clouds at Kuopio (a), Gual Pahari (b) and Elandsfontein (c). Horizontal line in box: median. Boxes: the upper and lower quartile. Whisker: extreme values.

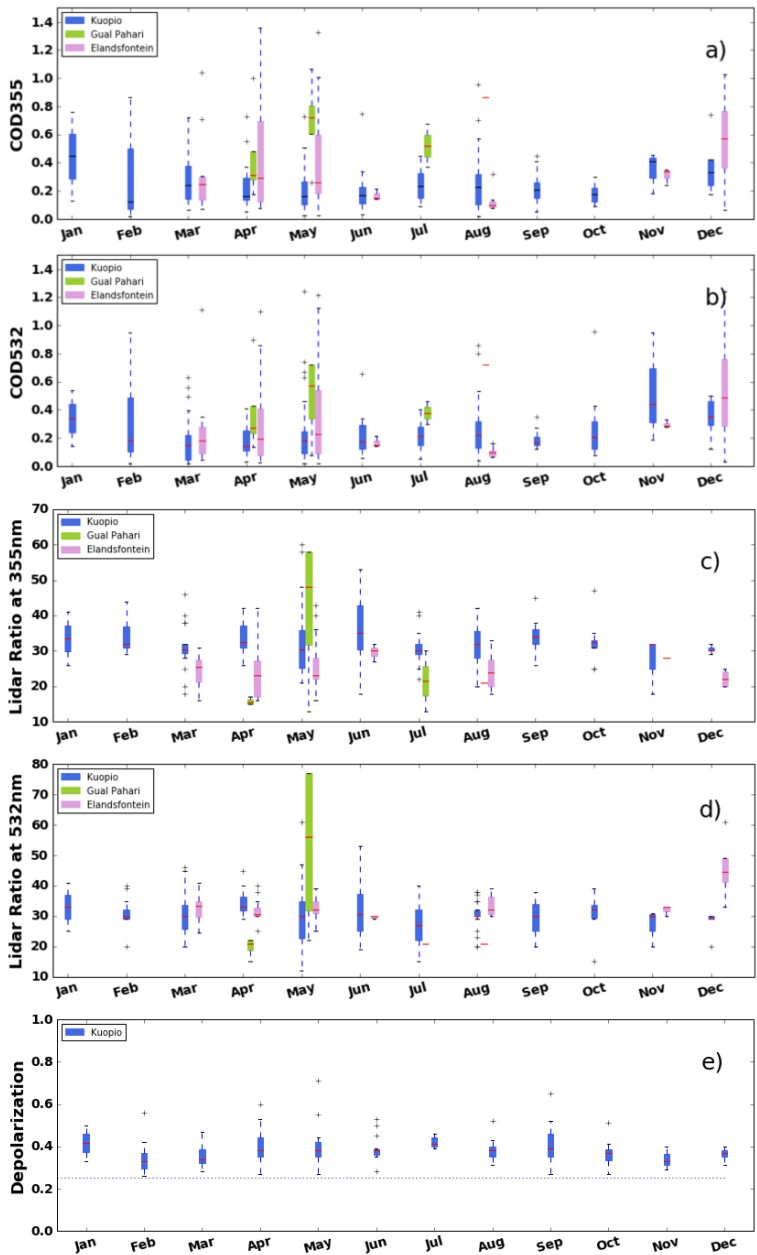

**Figure 6.** (a) Mean optical depth (multiple scattering corrected) values at 355nm, (b) mean optical depth (multiple scattering corrected) values at 532nm, (c) Lidar ratio at 355nm, (d) Lidar ratio at 532nm and (e) particle depolarization ratio for the detected cirrus layers for the study period of the three regions. Horizontal line in box: median. Boxes: the upper and lower quartile. Whisker: extreme values. Red line stands for the mean values for every month.

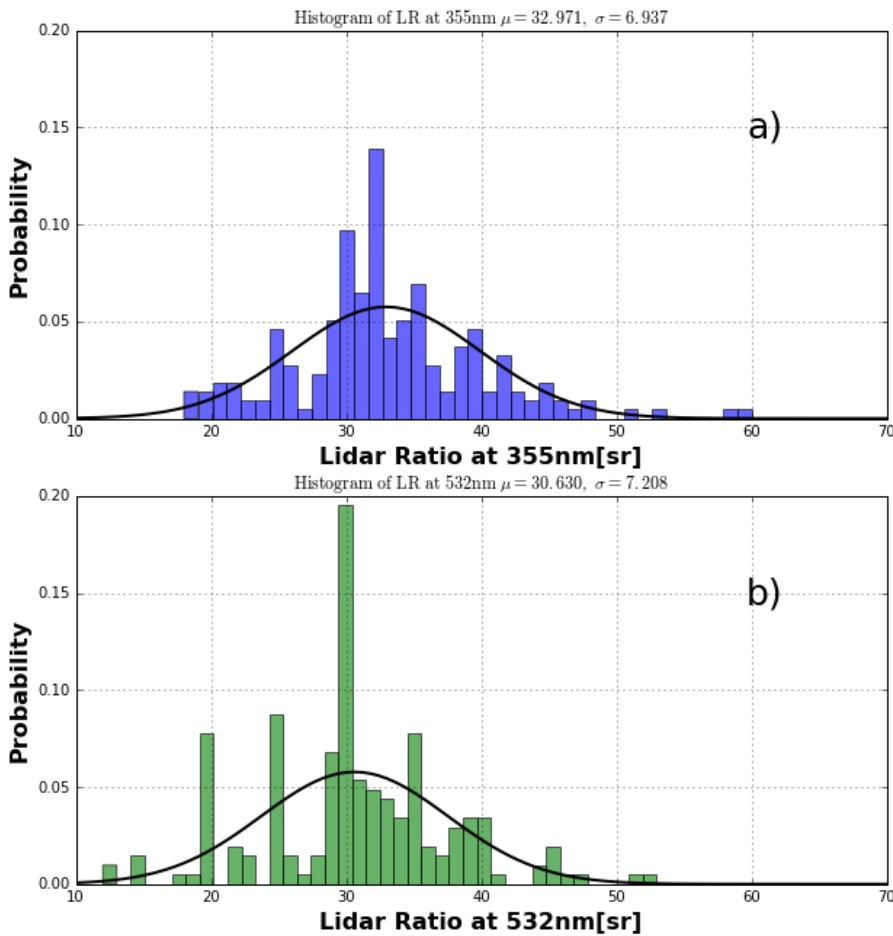

**Figure 7.** Histograms of (a) the Lidar ratio at 355nm and (b) the Lidar ratio at 532nm of the cirrus detected layers observed over Kuopio, Finland.

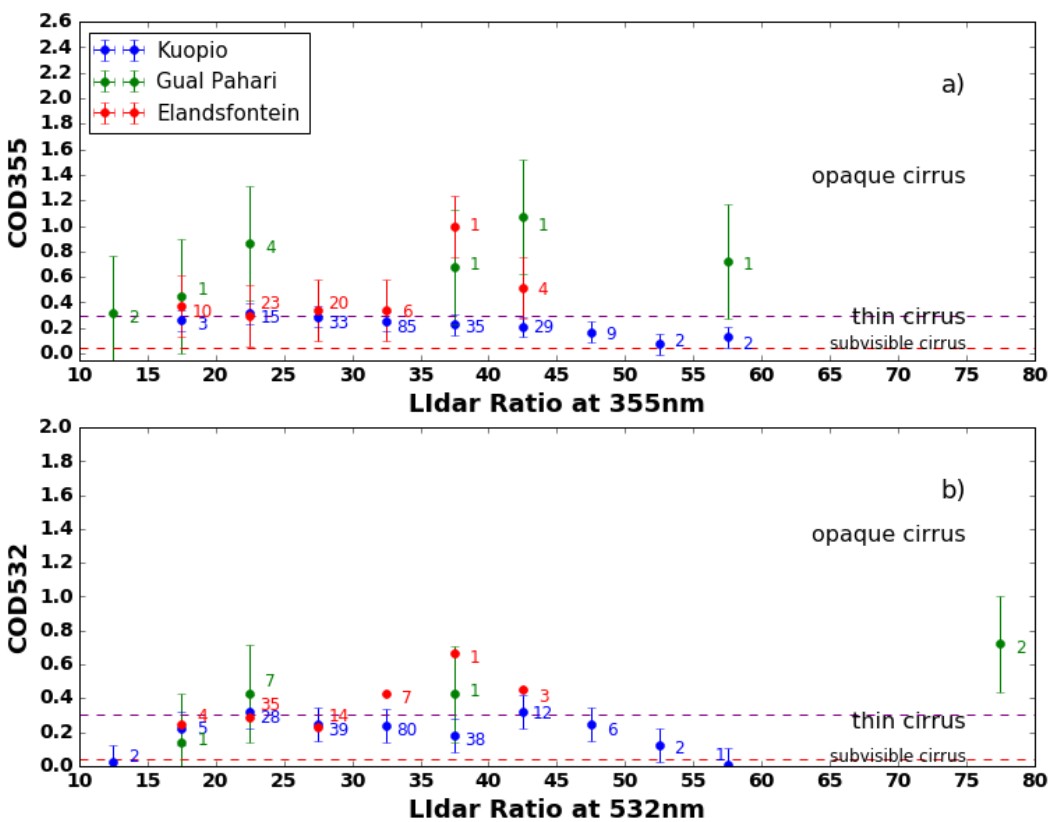

**Figure 8.** Dependence of (a) the corrected lidar ratio with COD at 355nm and (b) the corrected lidar ratio with COD at 532nm. Numbers labeled indicate the number of cases per lidar ratio bin. Horizontal dashed lines: cirrus categories by Sassen and Cho (1992).

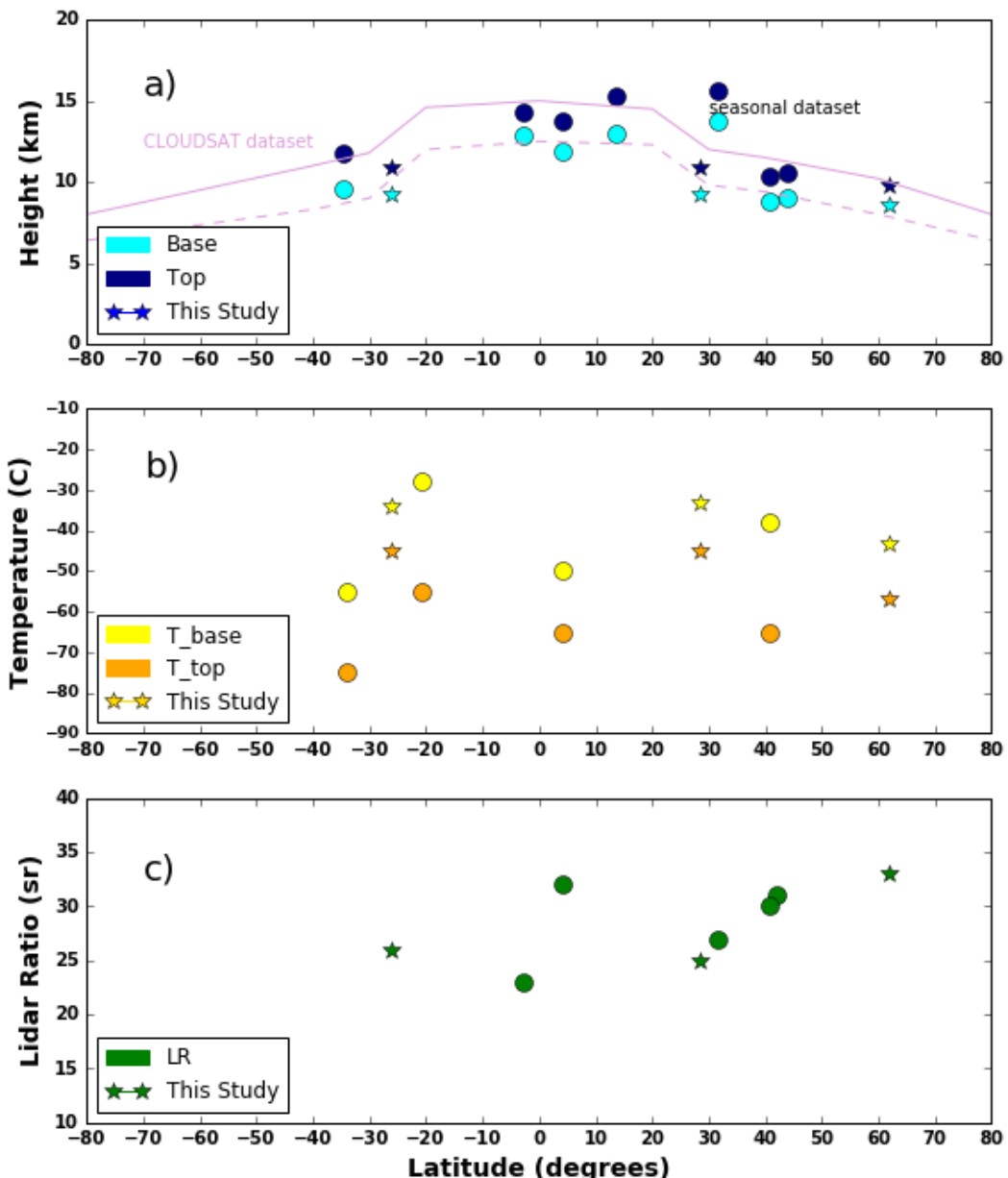

**Figure 9.** Latitudinal dependence of cirrus base and top height. Circles denote estimations from groundbased lidar from the literature (see Table 5 for references), stars denote estimations from this study and lines correspond to CLOUDSAT estimations according to Sassen et al. (2008) (a), latitudinal dependence of cirrus temperature base and top. Circles denote estimations from groundbased lidar from the literature (see Table 5 for references), stars denote estimations from this study and (b) same as above, but for latitudinal dependence of lidar ratio values (c).

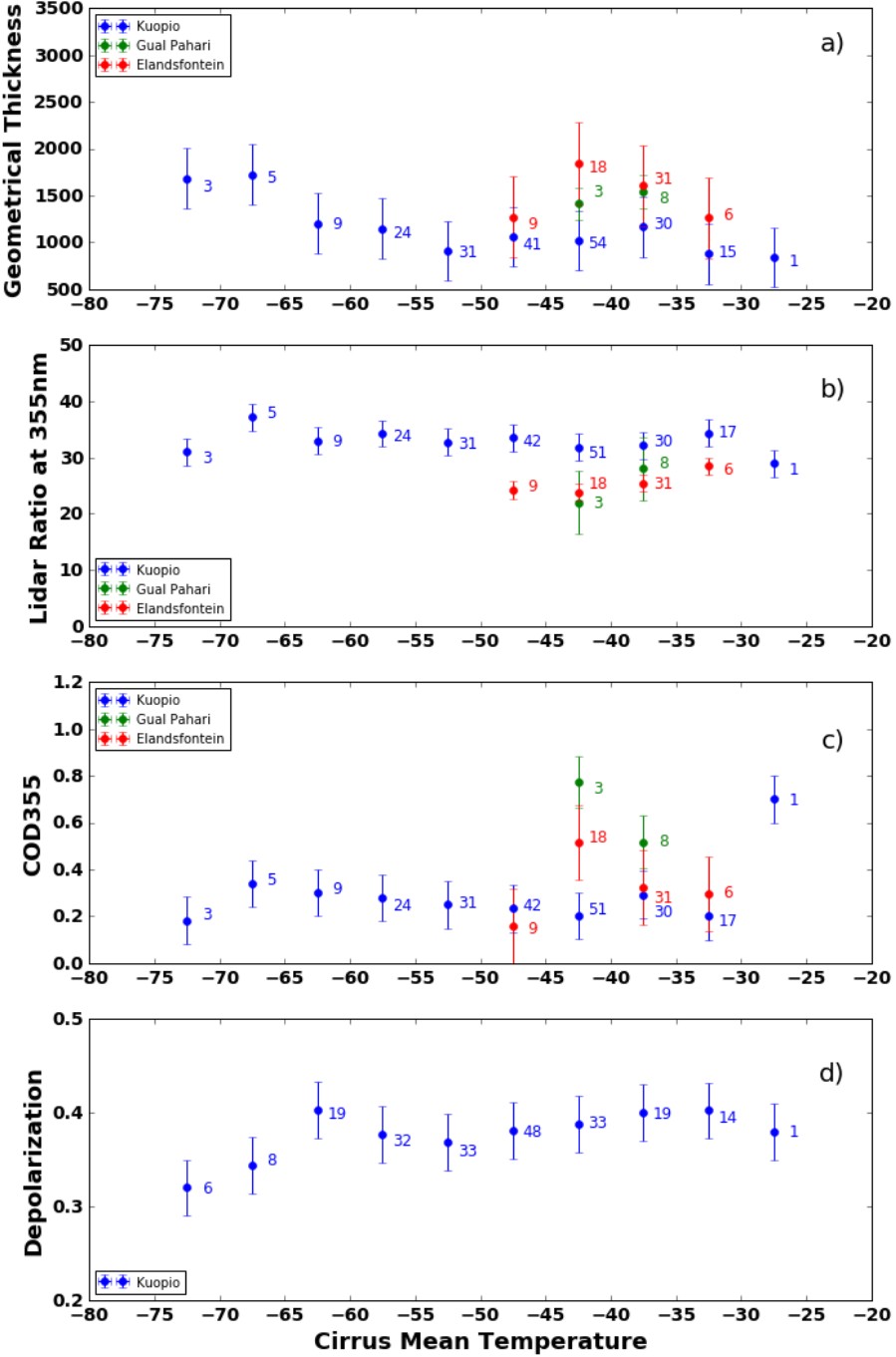

**Figure 10.** Dependencies of (a) geometrical thickness, (b) lidar ratio, (c) optical depth at 355nm and (d) particle depolarization values on 5°C intervals of cirrus mid temperature. Numbers labeled indicate the number of cases per temperature bin.

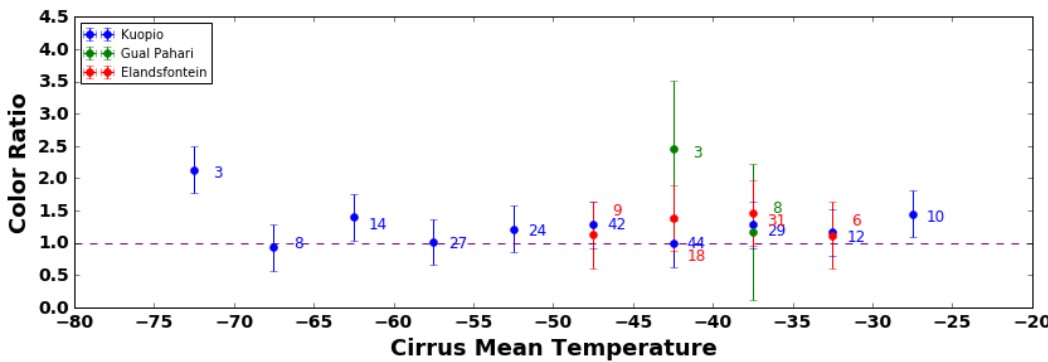

**Figure 11.** Dependencies of color ratio (355/532) on 5°C intervals of cirrus mid temperature. Numbers labeled indicate the number of cases per temperature bin.

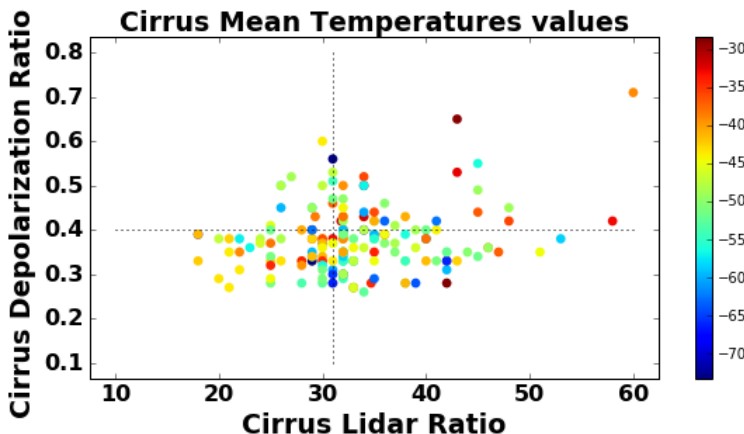

**Figure 12.** Dependencies of the mean temperature with the Lidar ratio values at 355nm and the particle depolarization values.