# Peer review of "Variability of cirrus cloud properties using a $Polly^{XT}$ Raman Lidar over high and tropical latitudes."

_Atmospheric Chemistry and Physics, 2019_

## Referee Comment (RC1) · Anonymous Referee #1 · 6 Sep 2019

General comments to acp-2019-565:

The paper presents the results of the cirrus cloud observations performed with a ground-based multi-wavelength PollyXT Raman Lidar during sequential periods between 2008 and 2016, in two subtropical stations (i.e. Gual Pahari in India and Elandsfontein in South Africa) and one subartic station (i.e. Kuopio in Finland). An automatic cirrus cloud detection algorithm was developed to derive the cirrus cloud lidar geometrical characteristics (cloud boundaries, geometrical thickness) and optical properties (cloud optical depth, lidar ratio, ice crystal depolarization ratio). Then, a statistical analysis and the seasonal variability of these parameters are presented comparing

the results of the three sites at different latitudes. The main results of the study are of interest. However, the authors should better characterize these results explaining the scientific context and their novelty and relevancy, re-organizing the structure of the paper that is not well structured. Some sections and figures lack of an accurate description and need to be completed. A more accurate characterization of the developed algorithm used to derive cirrus geometrical and optical properties is required, adding examples and/or references. The discussion of the results, which, in some parts, does not follow a linear path, should be modified giving more emphasis to the comparison among the different stations. The relationships between aerosol load and cirrus optical properties for the three different sites should be considered and discussed in the paper. Furthermore, an added value of this work could be providing an example on how and which the estimated cirrus parameters could be used in the parameterization schemes of the satellite optical retrievals. These issues need to be addressed to better present and to significantly strengthen the results. Thus, I recommend the publication of the manuscript after major revisions, according with the following observations.

Major comments:

Introduction

- The introduction lacks of a discussion about cirrus retrievals through CALIOP. Please add some discussions and references.

- Lines 61-67, the novelty of the work needs to be discussed and detailed.

Section 2

- Line 89, please add some details about the nature of the aerosols and their seasonality over the three different sites with appropriate references.

Section 3 and Section 4

As the retrieval of optical properties is part of the retrieval algorithm, I would suggest merging Section 3 and Section 4 in three different subsections (3.1, 3.2 and 3.3). The

contribution of the background aerosol load to the computation of cirrus cloud products is not negligible. This aspect should be discussed with the appropriate references not only in the last part of the Conclusions but also in this section. Please discuss this point.

- Lines 100-102, please discuss the normalization of signal (step b). How this normalization enhances the method applicability in different atmospheric conditions?

- Lines 113-115, it is not clear to me if the estimated cirrus geometrical and optical parameters are referred to 60-min averages. If yes, it means that, for example, for the cirrus of Fig.2 the detection algorithm will retrieve one cirrus parameter per hour. Is it correct? Please clarify this aspect.

- Lines 116- 119, please add more details and references about the criteria a) and c).

- Lines 143-144, the depolarization condition (particle linear depolarization > 0.25) is used only for Kuopio or also for the other sites? Which is the magnitude of the error/bias introduced by the Rayleigh calibration method? Despite the different calibration method, it could be of interest to show the depolarization ratio values of the other two sites.

- Line 156, it might be helpful to clarify the use of the Eloranta model writing the equation of the term $P_1(z)$ of the equation (4) and discussing the assumptions.

Section 5

The authors decided to present the results for the estimated geometrical and optical cirrus parameters for each site (sub-sections 5.0.2, 5.0.3 and 5.0.4, respectively). In my opinion, this choice makes the discussion of the results confusing. Another choice, which could help the comparison between subtropical and sub-artic sites, could be to divide the results according to geometrical and optical parameters (two sub-sections). This latter option allows both to improve the description and analysis of Fig. 4, Fig.6 and Fig. 7, where the parameters are depicted for all the stations, and to better compare each site. Table 2 should be completed adding also the value of all the other relevant cirrus parameters (e.g. mean/base/top heights, COD, temperature).

- Line 172, from Fig. 3 the diurnally variations cannot be observed. Please remove 'diurnally' or clarify.

- Lines 172-176, to analyze if the observed cirrus cover annual pattern is significant, it could be useful to show the number of total measurements per months. Considering your dataset, can you exclude that the observed cirrus cover annual pattern is only an indication of the annual pattern of low clouds/rain? Have you tried to compare this pattern with CALIOP observation over Kuopio region?

- Lines 176-177, could you explain the agreement between cirrus cover and temperature annual pattern? Are there similar results in literature?

- Lines 177-179, please add some numbers about the daytime/nighttime cirrus frequency and the number of total measurements. Could you explain these results?

- Lines 199-201, is this information relevant?

- Line 204, the AOD is referred to the column below the cirrus? Please explain. It could be of interest to relate AOD to cirrus parameters. Could you deepen this aspect?

- Lines 204-206 and 229-231, the discussion about Fig.5 is limited to these lines and does not give any relevant element of interest. Furthermore, concerning COD distribution, Sassen and Cho classification provides similar information. Please add some more elements of discussion or remove Fig. 5.

- Lines 216-229, the plot (e) of Fig. 6 is not discussed in the text. Please add some comments. The particle depolarization ratio together with LR and T could help to understand the cirrus crystal composition, size and shape. Did you find some relationship between delta and LR? Please add some comments and, if relevant, some results.

- Line 245, see comment of line 204.

- Line 262, see comment of line 204.

- Lines 265-269, please add in the discussion the results of the paper of Hoareau et al, 2013, about cirrus measurements at La Reunion sub-tropic site.

- Lines 300-323, this sub-section is of interest and it should be extended. In particular, on the basis of your analyses, is it possible to identify the parameters and the threshold values that could be used in satellite parameterization schemes? How the latitude dependence affect the variability of these parameters?

- Line 307, as already mentioned, the relationships between aerosol load and cirrus optical properties should be discussed more in details with a dedicated sub-section. In particular, the aerosol extinction below the cirrus and the type of aerosol could be of interest to understand the role of aerosol in cirrus formation. Do you have any analysis related to this?

- Line 310, could you explain the choice of using the cirrus base temperature instead of the mean/top temperature as independent parameter?

- Lines 315 and 319, please replace 'Fig.10' with 'Fig. 8'.

- Line 321, from Fig.8d the particle depolarization increases is not clear. Is it significant?

Conclusions

To summarize the results of this work, it would be useful to add a resuming table that, according to the different latitudes, compares the retrieved cirrus parameters to the main results of the literature reported in the paper.

---

## Referee Comment (RC2) · Anonymous Referee #2 · 30 Sep 2019

In the study discussed below, the authors around Kalliopi Artemis Voudouri present a cirrus cloud statistics for several sites where a multi-wavelength Raman lidar system of type Polly-XT was deployed.

The lidar data was analyzed with a newly developed cirrus retrieval algorithm. Cloud boundary detection was based on wavelet transformation of a kind of normalized background-corrected raw signals. For the identified cirrus layers cloud optical properties for 355-nm and 532-nm wavelength were derived and a multiple-scattering correction was applied.

The derived statistics of geometrical and optical cirrus cloud properties are diverse.

Clear conclusions could not be drawn.

I see a certain strengths in the manuscript, given the following facts:

- Presentation of a newly developed cirrus cloud identification algorithm

- Retrieval of Raman-based cirrus optical properties for 355 AND 532 nm

- Demonstration of the potential of a network of similar Raman lidar systems for application of one single retrieval scheme

Nevertheless, flaws in the description of the data analysis technique and in the discussion of the statistics dominate my impression while I was reading (several times) through the manuscript. I felt uncomfortable reading through the results section without knowing exactly how the statistics were derived. "Cases" are presented, but how is one case defined? To how many cloud profiles was the wavelet transform applied to get the boundaries? Are the presented optical properties based on Raman or Klett? These are important questions. Without knowledge about these, the value of the study is very limited.

I thus recommend a major revision of the manuscript, including a second review phase in order to put the study on a more solid footing.

Major comments:

1 – Ch.3; Retrieval Scheme/Fig. 1:

- The presentation of a retrieval scheme should always be done in such a way that others are able to reproduce it. The scheme given in Ch. 3 does not allow for that, because important information is missing:

o Was there range-averaging applied? If yes, under which conditions (see e.g. Fig. 1 b vs. 1 c)

o What happened to 1-hour intervals not filled entirely by cirrus clouds?

[Figure]

o How were irregularities in the cirrus cloud structure within the 1-hour averaging period treated?

o "the signal is normalized with a maximum value below 1.5km". What does this mean? Was the signal normalized using the maximum value found between the ground and 1.5 km height (or range)?

o In Eq. (1): What is z? altitude or range? Is this z the same z as the one in Eq. 3? I fear that range and height are mixed-up somewhat. Introduce separate variables for height and range where applicable/needed. Also: Is altitude above ground or above sea level (asl)? Was this considered in the statistics (given that the station elevation varies from 190 to 1745 m asl)?

o Eq. 2: What is Csig? Raw signal? Counts? What is Cbg? What is the difference between C and P (Eq. 1 vs. Eq. 2)?

o Case study/Fig. 2: The case study spans over about 4 hours, but the standard averaging period to derive the wavelet and particle depolarization ratio was 1 hour, wasn't it? I propose to show a case study that uses the actual time- and range-resolution used in the cirrus retrieval scheme.

o According the Figure 1, zero and background levels as well as normalization were applied to the range-corrected signal. Is this true? Shouldn't at least the background and zero values be subtracted from the raw signal?

o The selected base temperature of <-20°C (see Fig. 1) gives risk to the inclusion of layers of supercooled liquid water into the statistics, as ice formation occurs predominantly via the liquid phase at T>-27°C (Westbrook et al., 2011). Was there any threshold put on the temperature at cloud top? I'd believe a good value for this could be -38 °C or so in order to assure that at least at cloud top no liquid water was present any more.

o In Fig. 1: Again, what is altitude? Above sea level?

o In Fig. 1: Particle linear depolarization ratio is used as criteria for cirrus classification. But this parameter requires the detection of particle backscatter coefficient first. Shouldn't Fig. 1 thus contain an additional column (between CWT and cirrus criteria) that describes the calculation of the optical properties and multiple-scattering correction?

o In Fig. 2: Why do the cloud boundaries differ between (b) and (c)? Was there vertical smoothing applied to (c)?

2- Cirrus optical properties:

- During daytime, Klett-Fernald was applied, and during nighttime Raman was applied? Which values went into the statistics of lidar ratio, optical depth and particle depolarization ratio? Both? Only nighttime?

- How were reference height and values determined/set?

- In the results section, there should be a discussion of Klett-vs-Raman-based results.

3- Ch 4.02 Multiple Scattering correction:

- The lidar observations provide Ptot, but P1 is required. Eq. 4 thus contains 2 unknowns: P1, and F(z). How could the authors solve this equation?

4 – Ch. 5.01 Cirrus cloud cover detection:

- How is a case defined? What does it mean if there were 28 cases observed over Kuopio in April (P7, L175)?

- Table 2: What is N? Are these the number of hourly samples?

5 – Ch. 5.05:

- The title can be modified to 'cirrus classification at Kuopio' because the section only deals with this site.

6 – Ch. 5.06, Line 321:

- Could the decrease of particle LDR with increasing temperature be explained by the sporadic presence of supercooled liquid water?

7 - Conclusions:

- What conclusion can be drawn on the conversion of cirrus optical properties from 532 nm to 355 nm, considering that such conversion factors might be required to make future 355-nm/532-nm spaceborne lidar observations comperable? Can the authors make suggestions on which aspects future studies should look in more detail?

- In addition to the points addressed above, I recommend a thorough peer-review of spelling and grammar by the co-authors in beforehand to the submission of the revised manuscript.

- Minor comments will be addressed in the revised version.

References:

Westbrook, C. D., and Illingworth, A. J. ( 2011), Evidence that ice forms primarily in supercooled liquid clouds at temperatures $> -27°$C, Geophys. Res. Lett., 38, L14808, doi:10.1029/2011GL048021.

---

## Author Comment (AC1) · 20 Dec 2019

*The paper presents the results of the cirrus cloud observations performed with a ground-based multi-wavelength PollyXT Raman Lidar during sequential periods between 2008 and 2016, in two subtropical stations (i.e. Gual Pahari in India and Elandsfontein in South Africa) and one subartic station (i.e. Kuopio in Finland). An automatic cirrus cloud detection algorithm was developed to derive the cirrus cloud lidar geometrical characteristics (cloud boundaries, geometrical thickness) and optical properties (cloud optical depth, lidar ratio, ice crystal depolarization ratio). Then, a statistical analysis and the seasonal variability of these parameters are presented comparing the results of the three sites at different latitudes. The main results of the study are of interest. However, the authors should better characterize these results explaining the scientific context and their novelty and relevancy, re-organizing the structure of the paper that is not well structured. Some sections and figures lack of an accurate description and need to be completed. A more accurate characterization of the developed algorithm used to derive cirrus geometrical and optical properties is required, adding examples and/or references. The discussion of the results, which, in some parts, does not follow a linear path, should be modified giving more emphasis to the comparison among the different stations. The relationships between aerosol load and cirrus optical properties for the three different sites should be considered and discussed in the paper. Furthermore, an added value of this work could be providing an example on how and which the estimated cirrus parameters could be used in the parameterization schemes of the satellite optical retrievals. These issues need to be addressed to better present and to significantly strengthen the results. Thus, I recommend the publication of the manuscript after major revisions, according with the following observations.*

We thank the reviewer for his/her remarks that helped us to improve the manuscript. The reviewer is right that parts of the paper should be restructured, as in the present form is not easy to follow the comparisons of the cirrus properties between the different sites. In the revised version the reviewer's comments have been extensively taken into account, by improving the discussion of many sections (i.e., algorithm description, comparison among the different stations, adding new figures and tables) and by improving the figures that lacked of an accurate description. Below we report the changes included in the revised manuscript as a response to the comments of the reviewer.

*Introduction*

*1) The introduction lacks of a discussion about cirrus retrievals through CALIOP. Please add some discussions and references.*

The reviewer is right. The following text has been has been added in the revised version.

In the introduction section at page 2-3, we added the following paragraph:

[revised manuscript text omitted]

*Section 3 and Section 4*
*4) As the retrieval of optical properties is part of the retrieval algorithm, I would suggest merging Section 3 and Section 4 in three different subsections (3.1, 3.2 and 3.3). The contribution of the background aerosol load to the computation of cirrus cloud products is not negligible. This aspect should be discussed with the appropriate references not only in the last part of the Conclusions but also in this section. Please discuss this point.*

Section 3 and 4 have been merged accordingly. See also comment 3 for the discussion of the background aerosol load. In principal, we assume aerosol free (molecular) region in the altitudes above 6km where our algorithm is applied. This is a reasonable assumption. Climatological studies for these areas, show for Elandsfontein that the upper aerosol layers were found ~3.5km (Giannakaki et al., 2016), while for Gual Pahari the upper layers were below 6km (Komppula et al., 2012) for the period reported in the manuscript.

*5) Lines 100-102, please discuss the normalization of signal (step b). How this normalization enhances the method applicability in different atmospheric conditions?*

The normalization is applied to ensure the applicability of the method (use of threshold criteria for cirrus boundaries) to all the lidar systems. Given that lidar signals are uncalibrated and signal levels from one lidar system to another can be rather different, the normalization ensures the applicability of the criteria used by Baars et al., 2008. We normalized the range-corrected signal by its maximum value found below 1500 m. (below 2500 for Elandsfontein), which is usually the maximum value of the range corrected signal within the Boundary Layer, as proposed by Baars (2008), in order to use the same threshold values for the cirrus boundaries.

Baars, H., Ansmann, A., Engelmann, R., and Althausen, D.: Continuous monitoring of the boundary-layer top with lidar, Atmos. Chem. Phys., 8, 7281–7296, https://doi.org/10.5194/acp-8-7281-2008, 2008.

*6) Lines 113-115, it is not clear to me if the estimated cirrus geometrical and optical parameters are referred to 60-min averages. If yes, it means that, for example, for the cirrus of Fig.2 the detection algorithm will retrieve one cirrus parameter per hour. Is it correct? Please clarify this aspect.*

The reviewer is right. The estimated cirrus geometrical and optical parameters are referred to 60-min averages, so the Figure 3 has changed in the revised version of the manuscript, presenting the hourly means profiles.

*7) Lines 116- 119, please add more details and references about the criteria a) and c).*

The reviewer is right. We also applied an additional one, a threshold temperature to the cirrus top in order to assure that at cloud top no liquid water is present any more and we also changed the one in the cirrus base, so as to enhance the assumption that no liquid is present, as ice formation occurs predominantly via the liquid phase at T<-27°C (Westbrook et al., 2011).
The sentence has been changed to the following:

"Finally, cloud retrievals from the algorithm are classified as cirrus clouds when the following four criteria were met: i) the particle linear depolarization value is higher than 0.25 (Chen at al., 2002; Noel et al., 2002), ii) the altitude is higher than 6km and iii) the base temperature is below -27°C (Goldfarb et al., 2001; Westbrook et al., 2011) and iv) the top temperature is below -37°C (Campbell et al., 2015)."

(iv) the measured single scattering extinction profile (or the lidar ratio multiplied by the backscatter for the daytime measurements).

(v) the order of scattering

To calculate the multiple scattering correction, the code applies an iterative method including the following steps:

i) The measured extinction profile of the cirrus layer is provided.

ii) With the provided effective radius profile of the cirrus layer (linear relation of the effective radius with the cirrus temperature derived from radio soundings) and the effective (measured) extinction coefficient $\alpha$ par $(z)$, the model provides the ratio $P(z)/P(1)(z)$.

iii) From (2) a first value for the correcting factor $F(z)$ can be worked out.

iv) The iterative procedure continues till the calculation of a stable correcting factor $F(z)$ is found.

v) The corrected extinction can be then calculated from equation (5) in the manuscript and hence the value of lidar ratio.

The following sentence has been added in the manuscript: "The model assumes cirrus made up of hexagonal ice crystals and the model inputs are: (i) the laser beam divergence, (ii) the receiver field of view, (iii) the cirrus effective radius, (iv) the measured single scattering extinction profile (or the lidar ratio multiplied by the backscatter for the daytime measurements) and (v) the order of scattering. The estimation of the cirrus effective radius was taken from Wang and Sassen (2002), based on the linear relation of the effective radius with the cirrus cloud temperature derived from radio soundings.

For the multiple scattering calculation, the code applies an iterative method including the following steps:

i) The measured extinction profile of the cirrus layer is provided (a1).

ii) With the provided effective radius profile of the cirrus layer (linear relation of the effective radius with the cirrus temperature derived from radio soundings) and the measured extinction coefficient, an iterative procedure provides the ratio $P$ tot $(z)/P(z)$.

iii) From (ii) a first value for the correcting factor $F(z)$ can be worked out.

iv) The iterative procedure continues till the calculation of a stable correcting factor $F(z)$ is found.

v) The corrected extinction can be then calculated from equation (5) in the manuscript and hence the value of lidar ratio."

*Section 5*

10) *The authors decided to present the results for the estimated geometrical and optical cirrus parameters for each site (sub-sections 5.0.2, 5.0.3 and 5.0.4, respectively). In my opinion, this choice makes the discussion of the results confusing. Another choice, which could help the comparison between subtropical and sub-artic sites, could be to divide the results according to geometrical and optical parameters (two sub-sections). This latter option allows both to improve the description and analysis of Fig. 4, Fig.6 and Fig. 7, where the parameters are depicted for all the stations, and to better compare each site. Table 2 should be completed adding also the value of all the other relevant cirrus parameters (e.g. mean/base/top heights, COD, temperature).*

The reviewer is right. In the revised version of the manuscript the results are presented according to geometrical and optical parameters and Table 2 and Table 3 have been added with values of the other relevant cirrus properties. The discussion of the section is presented accordingly.

*11) Line 172, from Fig. 3 the diurnally variations cannot be observed. Please remove 'diurnally' or clarify.*

The reviewer is right. The word 'diurnally' has been removed in the revised version of the manuscript.

*12) Lines 172-176, to analyze if the observed cirrus cover annual pattern is significant, it could be useful to show the number of total measurements per months. Considering your dataset, can you exclude that the observed cirrus cover annual pattern is only an indication of the annual pattern of low clouds/rain? Have you tried to compare this pattern with CALIOP observation over Kuopio region?*

Figure shows the pattern of cirrus detection and not the pattern of actual cirrus occurrence. As Polly$^{XT}$ measures continuously (24/7) under favor weather conditions, indeed the pattern presented here is be strongly biased by the presence of low clouds and rain. Thus this pattern is not comparable with the one estimated from satellite retrievals and is quite different (see Figure 3 in Sassen et al, 2008).

*13) Lines 176-177, could you explain the agreement between cirrus cover and temperature annual pattern? Are there similar results in literature?*

The sentence has been rephrased, as it was quite misleading in the previous version of the manuscript. We did not aim to relate here cirrus cover with the annual pattern of temperature. Our purpose was to relate the low water clouds with temperature values and not the cirrus presence, in order to comment on the absence of cirrus measurements due to low clouds. So, now the sentence has been replaced with the following one:

"This monthly pattern of low clouds existence seems to follow the annual temperature cycle over the region (Jylhä et al., 2004), with maximum temperature values observed during the period April to October, while November to February are the coldest months."

*14) Lines 177-179, please add some numbers about the daytime/nighttime cirrus frequency and the number of total measurements. Could you explain these results?*

Table 2 in the revised version of the manuscript, shows the average cloud base and top altitudes and the average geometrical thickness for each site separating daytime and nighttime measurements. The averaged geometrical properties are found to be nearly identical above all sites, with differences less than 0.2km. Table 3 shows the averaged lidar ratio values, which found to be nearly identical above all except Gual Pahari site where average nightttime LR is 4sr higher than that of daytime.

"Table 2 summarizes the mean geometrical values calculated for each site. Differences between the mean values of the geometrical properties in the daytime and nighttime measurements are less than 200m for all sites."

"Table 3 summarizes the mean optical values discussed above, for the three sites, separating daytime and nighttime observations. Generally, the averaged optical properties values are found to be nearly identical, except one site (New Delhi), where average nighttime optical properties found higher than that of daytime. But since this dataset is limited, it cannot be used as a reference one."

*15) Lines 199-201, is this information relevant?*

The reviewer is right. The sentence has been removed in the revised version of the manuscript.

*16) Line 204, the AOD is referred to the column below the cirrus? Please explain. It could be of interest to relate AOD to cirrus parameters. Could you deepen this aspect?*

We apologize, this was a typo (AOD instead of COD). The correlation between AOD and cirrus properties would be a wholly different study. However, we proceed in the calculation of the AOD in the free troposphere, in order to have an indication about the calculated COD and the AOD below cirrus (check the answer to question 23).

*17) Lines 204-206 and 229-231, the discussion about Fig.5 is limited to these lines and does not give any relevant element of interest. Furthermore, concerning COD distribution, Sassen and Cho classification provides similar information. Please add some more elements of discussion or remove Fig. 5.*

We choose to keep the Figure 7 only with LR values plotted on and to comment accordingly in the results section. The figure is an evidence that although the lidar dataset is not continuous (due to unfavorable weather conditions during winter months), the frequency distributions are representative and even with this scarce sample of data, we observe consistent results with other literature studies.

So, we add a paragraph in the revised version of the manuscript:

"To further investigate the distribution of the cirrus lidar ratio values, we present a histogram of the values observed over Kuopio in Fig. 7. The most frequent measured lidar ratio values range between 28 and 36 sr for 355 nm and 20 and 36 sr for 532 nm. Similar results have been retrieved regarding the variability of LR 532, which is constant from one month to another, as shown. This figure can provide an evidence that although the lidar dataset are not continuous (due to unfavorable weather conditions duting winter months), the frequency distributions are close to normal and thus the statistics shown here have a significance. In addition we can claim that with this scarce sample of data we observe consistent results with a number of other literature studies."

*18) Lines 216-229, the plot (e) of Fig. 6 is not discussed in the text. Please add some comments. The particle depolarization ratio together with LR and T could help to understand the cirrus crystal composition, size and shape. Did you find some relationship between delta and LR? Please add some comments and, if relevant, some results.*

Indeed, the lidar ratio and the depolarization ratio are related to the microphysics and ice compositions of the cirrus clouds and to our knowledge no clear relationship is reported in literature. Concerning our retrievals, the following Figure (Figure 9 in the revised version of the manuscript) shows the relationship between the optical depth and the lidar ratio:

[Figure]

It can be seen from the above, that the highest values of Cirrus lidar ratio (>40) correspond to higher values of cirrus depolarization ratio (>0.4) and warmer cirrus. Moreover, it can be seen the variety of depolarization ratio values that correspond to the mean value of lidar ratio (~31). A similar behavior is reported in Chen et al. (2002) for lidar ratio values higher than 30 sr. In his study, the relationship between the depolarization ratio and the lidar ratios shows the former split into two groups for lidar ratios higher than 30. One group has high depolarization ratios about 0.5 and another group has 0.2.

The following sentence has been added in the revised version of the manuscript:
"The dependency of the mid temperature with the lidar ratio values at 355nm and the particle depolarization values is further examined. Figure 12 shows that the highest values of cirrus lidar ratio (>40) correspond to higher values of cirrus depolarization (>0.4) and warmer cirrus. Moreover, it can be seen the variety of depol values that correspond to the mean value of Lidar Ratio (∼ 31). A similar behavior is reported in Chen et al. (2002) for lidar ratio values higher than 30 sr. In his study, the relationship between the depolarization ratio and the lidar ratios shows the former split into two groups for lidar ratios higher than 30. One group has high depolarization ratios about 0.5 and another group has 0.2."

W. Chen, C. Chiang, and J. Nee, "Lidar ratio and depolarization ratio for cirrus clouds," Appl. Opt. 41, 6470-6476, 2002.

*19) Line 245, see comment of line 204.*
This is a typo, but see also comment on 23.

*20) Line 262, see comment of line 204.*
This is a typo, but see also comment on 23.

*21) Lines 265-269, please add in the discussion the results of the paper of Hoareau et al, 2013, about cirrus measurements at La Reunion sub-tropic site.*

The following sentence is added in the revised version of the manuscript:

"Our estimated thickness is slightly smaller than the reported mean value of 2km at La Reunion subtropical region (Hoareau et al., 2012). " Also the values of their study have been added in Table 5 and in Figure 9.

*Hoareau, C., Keckhut, P., Baray, J.-L., Robert, L., Courcoux, Y., Porteneuve, J., Vömel, H., and Morel, B.: A Raman lidar at La Reunion (20.8° S, 55.5° E) for monitoring water vapour and cirrus distributions in the subtropical upper troposphere: preliminary analyses and description of a future system, Atmos. Meas. Tech., 5, 1333–1348, https://doi.org/10.5194/amt-5-1333-2012, 2012.*

*22) Lines 300-323, this sub-section is of interest and it should be extended. In particular, on the basis of your analyses, is it possible to identify the parameters and the threshold values that could be used in satellite parameterization schemes? How the latitude dependence affect the variability of these parameters?*

As latitudinal dependence is found, different parameterization would be necessary to future satellite retrievals. The parameters to be taken into account should be:

i) the LR value selected for the different counterparts. A latitudinal variation can be seen in the lidar ratios with the lowest values found over the tropical region and over South Africa (Table 2 and Table 3). ii) LR values seem to be almost constant between day and night retrievals, except for the Gual Pahari site, where the dataset is not extensive and could not be used as a reference one.

Also Table 3 has been added, summarizing most of the cirrus clouds geometrical and optical properties of ground-based lidar observations reported in literature. Overall, we can conclude that cirrus cloud thicknesses are greatest in the tropics and decrease toward the poles and concerning the lidar ratio values, these seem to be an increasing trend towards the north pole.

The following sentence is added in the revised version of the manuscript:

"The reported values in literature from previous studies based on lidar ground-based datasets and the current one, are listed in Table 4. Generally, cirrus layers have been observed up to altitudes of 13km above Gual Pahari, whereas they have only been detected to about 1km lower at the other two regions and this conclusion is in accordance with the Cloudsat observations (Sassen et al., 2008). Based on the satellite information, the derived cirrus cloud thicknesses was found to be larger in the tropics and decreasing toward the poles. Also from the values reported from groundbased studies, a pattern can be concluded: cirrus cloud geometrical properties peaks around the equator and at midlatitudes sites, with generally decreasing amounts as the poles are approached. On the other hand, the lidar ratio values seem to follow a diverse relation, showing greater values moving to the poles. In our study, larger values of LR is found for the subarctic station and smaller LR were observed for Gual Pahari and Elandsfontein."

*23) Line 307, as already mentioned, the relationships between aerosol load and cirrus optical properties should be discussed more in details with a dedicated sub-section. In particular, the aerosol extinction below the cirrus*

*and the type of aerosol could be of interest to understand the role of aerosol in cirrus formation. Do you have any analysis related to this?*

| Measurement Site | Kuopio | Elandsfontein | Gual Pahari |
|---|---|---|---|
| Type of aerosols | Fine particles | biomass burning aerosols and mixtures of biomass burning aerosols with desert dust particles and urban/industrial particles | dust particles, biomass burning |
| AOD FT | 0.01 ± 0.01 | 0.06 ± 0.04 | 0.09 ± 0.03 |
| COD | 0.25 ± 0.27 | 0.35 ± 0.30 | 0.60 ± 0.40 |

Giannakaki et al. (2016) showed that the aerosol classification over South Africa indicates mostly biomass burning aerosols and mixtures of biomass burning aerosols with desert dust particles, as well as the possible continuous influence of urban/industrial aerosol load in the region.

New Delhi, represents a semi-urban environment surrounded mainly by agricul tural test fields and light vegetation. The seasonal characteristics of the aerosol vertical structure performed there (Komppula et al., 2012) indicate the presence of dust particles due to the Asian dust storms.

Kuopio is a semiurban site, where mostly fine particles exist (Aaltonen et al., 2012).

We thank the reviewer for this comment. Table 5 is added in the revised version of the manuscript, including all the information of the cirrus properties based on ground-based lidar observations reported in the literature for the different latitudes. A latitudinal dependence of the cirrus cloud can be seen. See also answer to question number 22.

The following sentence is added in the revised version of the manuscript:

"The three presented datasets are derived from different latitudinal and climatic sites. In this section we firstly examine the latitudinal dependence of the cirrus geometrical and optical properties. The reported values in literature from previous studies based on lidar grounbased dataset and the retrievals of the current one are listed in Table 5 and plotted in Figure 9 for comparison. We can note, that the cirrus geometrical properties and the lidar ratio values may vary greatly depending on the latitude and an decreasing trend of the geometrical boundaries with the rise of the distance from the equator is obvious, also reported by satellite observations (Sassen et al., 2008). Generally, cirrus layers have been observed up to altitudes of 13km above Gual Pahari, whereas they have only been detected to about 1km lower at the subarctic region and this conclusion is in accordance with the Cloudsat observations (Sassen et al., 2008). Based on the satellite information, the derived cirrus cloud thicknesses was found to be larger in the tropics and decreasing toward the poles. Also from the values reported from groundbased studies, a pattern can be concluded: cirrus cloud geometrical properties peaks around the equator and at midlatitudes sites, with generally decreasing amounts as the poles are approached. On the other hand, the optical properties seem to follow a diverse relation, showing greater values moving to the poles. In our study, larger values of COD 355 and smaller LR were observed for Gual Pahari and Elandsfontein. The larger variability of the optical properties at the two subtropical regions, relative to Kuopio, could be related to the larger and variable aerosol load over these regions."

---

## Author Response (AR1)

The paper presents the results of the cirrus cloud observations performed with a groundbased multi-wavelength PollyXT Raman Lidar during sequential periods between 2008 and 2016, in two subtropical stations (i.e. Gual Pahari in India and Elandsfontein in South Africa) and one subartic station (i.e. Kuopio in Finland). An automatic cirrus cloud detection algorithm was developed to derive the cirrus cloud lidar geometrical characteristics (cloud boundaries, geometrical thickness) and optical properties (cloud optical depth, lidar ratio, ice crystal depolarization ratio). Then, a statistical analysis and the seasonal variability of these parameters are presented comparing the results of the three sites at different latitudes. The main results of the study are of interest. However, the authors should better characterize these results explaining the scientific context and their novelty and relevancy, re-organizing the structure of the paper that is not well structured. Some sections and figures lack of an accurate description and need to be completed. A more accurate characterization of the developed algorithm used to derive cirrus geometrical and optical properties is required, adding examples and/or references. The discussion of the results, which, in some parts, does not follow a linear path, should be modified giving more emphasis to the comparison among the different stations. The relationships between aerosol load and cirrus optical properties for the three different sites should be considered and discussed in the paper. Furthermore, an added value of this work could be providing an example on how and which the estimated cirrus parameters could be used in the parameterization schemes of the satellite optical retrievals. These issues need to be addressed to better present and to significantly strengthen the results. Thus, I recommend the publication of the manuscript after major revisions, according with the following observations.

We thank the reviewer for his/her remarks that helped us to improve the manuscript. The reviewer is right that parts of the paper should be restructured, as in the present form is not easy to follow the comparisons of the cirrus properties between the different sites. In the revised version the reviewer's comments have been extensively taken into account, by improving the discussion of many sections (i.e., algorithm, comparison among the different stations) and by improving the figures that lacked of an accurate description. Below we report the changes included in the revised manuscript as a response to the comments of the reviewer.

Introduction

1) The introduction lacks of a discussion about cirrus retrievals through CALIOP. Please add some discussions and references.

The reviewer is right. The following text has been has been added in the revised version.

In the introduction section at page 2-3, we added the following paragraph:

"There are also satellite based studies from either lidar (CloudAerosol Lidar with Orthogonal Polarization (CALIOP), Dupont et al., 2010) or cloud radar (CloudSat) or combined lidar and cloud radar (e.g. Sassen et al. 2008) retrievals that provide a global view concerning the seasonal frequencies of cirrus clouds and their geometrical and optical properties and their variabilities.

However, comparison on cirrus cloud properties performed with ground-based lidar systems for long periods at different geographical locations are scarce, even though that kind of observations that correspond to different areas and atmospheric conditions are crucial to reveal information of the latitudinal dependence of the cirrus properties and indications about the aerosol effect on the geometrical and optical characteristics of the detected cirrus layers. That kind of observations can be further used in the validation of the satellite retrievals. which provide global distribution of cirrus clouds (Sassen et al., 2008). Given that for satellite retrievals, the main input parameter to the optical processing of the cirrus lavers is the lidar ratio, the selected lidar ratio value can introduce errors on the retrieved extinction and optical depth values of the cirrus layers, as it is illustrated by Young et al., (2018). The optical depth comparison of the Version 4.10 (V4) of the CALIOP optical depths and the optical depths reported by MODIS collection 6 show substantial improvements relative to earlier comparisons between CALIOP version 3 and MODIS collection 5, as a result of extensive upgrades of the extinction retrieval algorithm. New apriori information of the lidar ratio value for the cirrus layers, included in Version 4.10 (V4) of the CALIOP data products, led to improvements of the extinction and optical depth estimates of the cirrus cloud lavers. Thus, ground based lidar observations of the cirrus properties. that correspond to different areas and atmospheric conditions, are crucial to verify and eventually improve the satellite retrievals.

**Reference:**

Dupont, J.-C., M. Haeffelin, Y. Morille, V. Noël, P. Keckhut, D. Winker, J. Comstock, P. Chervet, and A. Roblin Macrophysical and optical properties of midlatitude cirrus clouds from four ground-based lidars and collocated CALIOP observations, J. Geophys. Res., 115, D00H24, doi:10.1029/2009JD011943, 2010.

Sassen, K., Z. Wang, and D. Liu, Global distribution of cirrus clouds from CloudSat/Cloud-Aerosol Lidar and Infrared Pathfinder Satellite Observations (CALIPSO) measurements, J. Geophys. Res., 113, D00A12, doi:10.1029/2008JD009972, 2008.

Young, S. A., Vaughan, M. A., Garnier, A., Tackett, J. L., Lambeth, J. D., and Powell, K. A.: Extinction and optical depth retrievals for CALIPSO's Version 4 data release, Atmos. Meas. Tech., 11, 5701–5727, https://doi.org/10.5194/amt-11-5701-2018, 2018.

2) Lines 61-67, the novelty of the work needs to be discussed and detailed.

The text is modified accordingly and the following sentence is added:

"Ground-based lidar data sets gathered over three sites, two subtropical and one subarctic, are used to evaluate the consistency of cloud geometrical and optical property climatologies that can be derived by such data sets. The aim of this work is firstly to analyze the cirrus properties over sites in different hemispheres, from observations derived with the same ground based lidar system and secondly to attribute any observable differences to the subarctic and subtropical counterparts. The cirrus properties differences are discussed in order to provide identifications of the possible causes of discrepancies. The study tries to give answers to the following questions:

- Is there a common pattern of the cirrus geometrical and optical properties over different latitudes?

- Can a groundbased lidar system provide.

The analysis presented here could further assist in bridging existing gaps relating to cirrus properties over different regions, since limited long-term cirrus characteristics are available. The information of the lidar ratio is an important parameter for the inversion of lidar signals in instruments that do not have Raman channel and space-based lidars, such as CALIPSO, depend on a parameterization that may vary with location. Thus, information provided by well calibrated ground based measurements is quite critical. Analysis of the lidar ratios values derived from lidar measurements in different parts of the world, where various atmospheric conditions exist will produce results that are more representative of the actual conditions and reductions in the uncertainties related to these lidar ratios will reduce the uncertainties in the retrieved extinction coefficients and derived optical depths."

**3)* Section 2, Line 89, please add some details about the nature of the aerosols and their seasonality over the three different sites with appropriate references.**

The text is modified accordingly and the following paragraph is added:

"The one year aerosol analysis of lidar observations in Gual Pahari (Komppula et al., 2012) showed that in the summer, the measured air masses were slightly more polluted and the particles were a bit larger than in other seasons (higher Angström exponent values), with the main aerosol sources to be the local and regional biomass and fossil fuel burning. The annual averages revealed a distinct seasonal pattern of aerosol profiles, with aerosol concentrations slightly higher in summer (June – August) compared to other seasons, and particles larger in size. During the summer and autumn, the average lidar ratios were larger than 50 sr, suggesting the presence/dominance of absorbing aerosols from biomass burning. The lidar observations that were performed at Elandsfontein and used for aerosol characterization for the corresponding study period (Giannakaki et al. 2016; Giannakaki et al. 2016) showed that the observed layers were classified as urban/industrial, biomass burning, and mixed aerosols using the information of backward trajectories, MODIS hotspot fire products and in situ aerosol observations. The analysis of the seasonal pattern of vertical profiles of the aerosol optical properties showed that the more absorbing (higher Lidar ratio at 355 nm) biomass particles were larger on August and October, while the category of Urban/industrial had their peak on January, March and May. Another study of the seasonal aerosol climatological characteristics, based on 10 years of MISR data (Tesfaye et al., 2011), showed that the aerosol extinction optical depth

values at 558 nm illustrate annual trends with a maximum during early summer (November–February) and minimum during winter (May–July). Generally, during summer and early winter, the area is dominated by a mixture of coarse-mode and accumulation-mode particles, while during winter and early summer, it is dominated by submi-cron particles.

Kuopio is an urban area and constitutes a low-aerosol-content environment. The columnar analysis of sunphotometer observations (Aaltonen et al., 2010) revealed that the high Angstrom exponent values observed can be possible linked with the presence of fine particles, while the seasonal analysis of the optical depth showed that there is no significant variation."

(iv) the measured single scattering extinction profile (or the lidar ratio multiplied by the backscatter for the daytime measurements).

(v) the order of scattering

To calculate the multiple scattering correction, the code applies an iterative method including the following steps:

i) The measured extinction profile of the cirrus layer is provided.

ii) With the provided effective radius profile of the cirrus layer (linear relation of the effective radius with the cirrus temperature derived from radio soundings) and the effective (measured) extinction coefficient  $\alpha$  par (z), an iterative procedure provides the ratio P (z)/P (1) (z).

iii) From (2) a first value for the correcting factor F(z) can be worked out

iv) The iterative procedure continues till the calculation of a stable correcting factor F(z) is found.

v) The corrected extinction can be then calculated from equation (5) in the manuscript and hence the value of lidar ratio.

The following sentence has been added in the manuscript: "The model assumes cirrus made up of hexagonal ice crystals and the model inputs are: (i) the laser beam divergence, (ii) the receiver field of view, (iii) the cirrus effective radius, (iv) the measured single scattering extinction profile (or the lidar ratio multiplied by the backscatter for the daytime measurements) and (v) the order of scattering. The estimation of the cirrus effective radius was taken from Wang and Sassen (2002), based on the linear relation of the effective radius with the cirrus cloud temperature derived from radio soundings."

**Section 5**

10) The authors decided to present the results for the estimated geometrical and optical cirrus parameters for each site (sub-sections 5.0.2, 5.0.3 and 5.0.4, respectively). In my opinion, this choice makes the discussion of the results confusing. Another choice, which could help the comparison between subtropical and sub-artic sites, could be to divide the results according to geometrical and optical parameters (two sub-sections). This latter option allows both to improve the description and analysis of Fig. 4, Fig.6 and Fig. 7, where the parameters are depicted for all the stations, and to better compare each site. Table 2 should be completed adding also the value of all the other relevant cirrus parameters (e.g. mean/base/top heights, COD, temperature).

The reviewer is right. In the revised version of the manuscript the results are presented according to geometrical and optical parameters and Table 2 and Table 3 have been added with values of the other relevant cirrus properties. The discussion of the section is presented accordingly.

11) Line 172, from Fig. 3 the diurnally variations cannot be observed. Please remove 'diurnally' or clarify.

The reviewer is right. The word 'diurnally' has been removed in the revised version of the manuscript.

12) Lines 172-176, to analyze if the observed cirrus cover annual pattern is significant, it could be useful to show the number of total measurements per months. Considering your dataset, can you exclude that the observed cirrus cover annual pattern is only an indication of the annual pattern of low clouds/rain? Have you tried to compare this pattern with CALIOP observation over Kuopio region?

Figure shows the pattern of cirrus detection and not the pattern of cirrus occurrence. As PollyXT measures continuously (24/7) under favor weather conditions, indeed the pattern presented here is only an indication and biased by the presence of low clouds and rain. Even though lidars provide information only in rain/low cloud free times, the pattern of seasonality is consistent with satellite retrievals. Concerning the Elandsfontein site, the average seasonal (monthly) dependence of cirrus cloud frequencies from the CLOUDSAT/CALIPSO observations presented by Sassen (2008), follows the same pattern presented in this study.

13) Lines 176-177, could you explain the agreement between cirrus cover and temperature annual pattern? Are there similar results in literature?

The sentence has been rephrased, as it was quite misleading in the previous version of the manuscript. Our purpose was to relate the low water clouds with temperature values and not the cirrus detection. So, now the sentence has been replaced with the following one:

"This monthly pattern of low clouds existence seems to follow the annual temperature cycle over the region (Jylhä et al., 2004), with maximum temperature values observed during the period April to October, while November to February are the coldest months."

14) Lines 177-179, please add some numbers about the daytime/nighttime cirrus frequency and the number of total measurements. Could you explain these results?

Table 3 in the revised version of the manuscript, 3 shows the number of detected cirrus layer per site and the number of daytime and nighttime measurements and the following sentence has been added:

The following sentences has been added in the revised version of the manuscript: "Table 2 summarizes the mean geometrical values calculated for each site. We can conclude that the differences between the mean values of the geometrical properties in the daytime and nighttime measurements are not statistically significant. "

"Table 3 summarizes the mean optical values discussed above, for the three sites, separating daytime and nighttime observations. Generally, the averaged optical properties values are found to be nearly identical, except one site (New Delhi), where average nighttime optical properties found higher than that of daytime."

**15) Lines 199-201, is this information relevant?**

The reviewer is right. The sentence has been removed in the revised version of the manuscript.

**16) Line 204, the AOD is referred to the column below the cirrus? Please explain. It could be of interest to relate AOD to cirrus parameters. Could you deepen this aspect?**

We apologize, this was a typo (AOD instead of COD). The correlation between AOD and cirrus properties would be a wholly different study. However, we proceed in the calculation of the AOD in the free troposphere, in order to have an indication about the calculated COD and the AOD below cirrus (check the answer to question 23).

17) Lines 204-206 and 229-231, the discussion about Fig.5 is limited to these lines and does not give any relevant element of interest. Furthermore, concerning COD distribution, Sassen and Cho classification provides similar information. Please add some more elements of discussion or remove Fig. 5.

We choose to keep the Figure 7 and to add comments in the results. This figure can provide an evidence that although the lidar dataset is not continuous (due to unfavor weather conditions during winter months), the frequency distributions are representative and even with this scarce sample of data we observe consistent results with other literature studies.

So, we add a paragraph in the revised version of the manuscript:

"To further investigate the distribution of the cirrus lidar ratio values over Kuopio, we present a histogram of the values derived in Fig. 7. The most frequent observed lidar ratio values ranging between 28 and 36 sr for 355 nm and 20 and 36 sr for 532 nm. Similar results have been retrieved regarding the variability of LR 532, which is constant from one month to another, as shown. This figure can provide an evidence that although the lidar dataset are not continuous (due to not favour weather conditions the winter months), the frequency distributions are close to normal and thus the statistics shown here have a significance. In addition we can claim that with this scarce sample of data we observe consistent results with a number of other literature studies."

18) Lines 216-229, the plot (e) of Fig. 6 is not discussed in the text. Please add some comments. The particle depolarization ratio together with LR and T could help to understand the cirrus crystal composition, size and shape. Did you find some relationship between delta and LR? Please add some comments and, if relevant, some results.

Indeed, the lidar ratio and the depolarization ratio are related to the microphysics and ice compositions of the cirrus clouds and to our knowledge no clear relatioship is reported in literature. Concerning our retrievals, the following Figure (Figure 9 in the revised version of the manuscript) shows the relationship between the optical depth and the lidar ratio:

It can be seen from the above, that the highest values of Cirrus lidar ratio (>40) correspond to higher values of Cirrus Depol (>0.4) and warmer cirrus. Moreover, it can be seen the variety of depol values that correspond to the mean value of lidar ratio ( $\sim$ 31). A similar behavior is reported in Chen et al. (2002).

The following sentece has been added in the revised version of the manuscript: "The dependency of the mid temperature with the lidar ratio values at 355nm and the particle depolarization values is further examined. Figure 12 shows that the highest values of cirrus lidar ratio (>40) correspond to higher values of cirrus depolarization (>0.4) and warmer cirrus. Moreover, it can be seen the variety of depol values that correspond to the mean value of Lidar Ratio (~ 31). A similar behavior is reported in Chen et al. (2002) for lidar ratio values higher than 30 sr. In his study, the relationship between the depolarization ratio and the lidar ratios shows the former split into two groups for lidar ratios higher than 30. One group has high depolarization ratios about 0.5 and another group has 0.2."

W. Chen, C. Chiang, and J. Nee, "Lidar ratio and depolarization ratio for cirrus clouds," Appl. Opt. 41, 6470-6476, 2002.

*19) Line 245, see comment of line 204.* This is a typo, but see also comment on 23.

20) Line 262, see comment of line 204. This is a typo, but see also comment on 23.

21) Lines 265-269, please add in the discussion the results of the paper of Hoareau et al, 2013, about cirrus measurements at La Reunion sub-tropic site.

The following sentence is added in the revised version of the manuscript:

"Our estimated thickness is slightly smaller than the reported mean value of 2km at La Reunion subtropical region (Hoareau et al., 2012)." Also the values of their study have been added in Table 5.

Hoareau, C., Keckhut, P., Baray, J.-L., Robert, L., Courcoux, Y., Porteneuve, J., Vömel, H., and Morel, B.: A Raman lidar at La Reunion (20.8° S, 55.5° E) for monitoring water vapour and cirrus distributions in the subtropical upper troposphere: preliminary analyses and description of a future system, Atmos. Meas. Tech., 5, 1333–1348, https://doi.org/10.5194/amt-5-1333-2012, 2012.

22) Lines 300-323, this sub-section is of interest and it should be extended. In particular, on the basis of your analyses, is it possible to identify the parameters and the threshold values that could be used in satellite parameterization schemes? How the latitude dependence affect the variability of these parameters?

As latitudinal dependence is found, different parameterization would be necessary to future satellite retrievals. The parameters to be taken into account should be:

i) the LR value selected for the different counterparts. A latitudinal variation can be seen in the lidar ratios with the lowest values found over the tropical region and over South Africa (Table 2 and Table 3).

ii) LR values seem to be almost constant between day and night retrievals, except for the Gual Pahari site, where the dataset is not extensive and could not be used as a reference one.

Also Table 3 has been added, summarizing most of the cirrus clouds geometrical and optical properties of ground-based lidar observations reported in literature. Overall, we can conlude that cirrus cloud thicknesses are greatest in the tropics and decrease toward the poles and concerning the lidar ratio values, these seem to be increasing towards the north pole.

The following sentence is added in the revised version of the manuscript:

"The reported values in literature from previous studies based on lidar grounbased dataset and the current one are listed in Table 4. Generally, cirrus layers have been observed up to altitudes of 13km above Gual Pahari, whereas they have only been detected to about 1km lower at the other two regions and this conclusion is in accordance with the Cloudsat observations (Sassen et al., 2008). Based on the satellite information, the derived cirrus cloud thicknesses was found to be larger in the tropics and decreasing toward the poles. Also from the values reported from groundbased studies, a pattern can be concluded: cirrus cloud geometrical properties peaks around the equator and at midlatitudes sites, with generally decreasing amounts as the poles are approached. On the other hand, the lidar ratio values seem to follow a diverse relation, showing greater values moving to the poles. In our study, larger values of LR is found for the subarctic station and smaller LR were observed for Gual Pahari and Elandsfontein."

23) Line 307, as already mentioned, the relationships between aerosol load and cirrus optical properties should be discussed more in details with a dedicated sub-section. In particular, the aerosol extinction below the cirrus and the type of aerosol could be of interest to understand the role of aerosol in cirrus formation. Do you have any analysis related to this?

| Measurement Site | Kuopio         | Elandsfontein                                                                                     | Gual Pahari                        |
|------------------|----------------|---------------------------------------------------------------------------------------------------|------------------------------------|
| Type of aerosols | Fine particles | biomass burning
aerosols and
mixtures of
biomass burning
aerosols with
desert dust | dust particles,
biomass burning |

|        |                    | particles and
urban/industrial
particles |                |
|--------|--------------------|------------------------------------------------|----------------|
| AOD FT | $0.0056 \pm 0.009$ | $0.058 \pm 0.04$                               | 0.0875 ± 0.034 |
| COD    | 0.25 ± 0.27        | 0.4 ± 0.3                                      | 0.59 ± 0.39    |

Giannakaki et al. (2016) showed that the aerosol classification over South Africa indicates mostly biomass burning aerosols and mixtures of biomass burning aerosols with desert dust particles, as well as the possible continuous influence of urban/industrial aerosol load in the region.

New Delhi, represents a semi-urban environment surrounded mainly by agricul tural test fields and light vegetation. The seasonal characteristics of the aerosol vertical structure performed there (Komppula et al., 2012) indicate the presence of dust particles due to the Asian dust storms.

Kuopio is a semiurban site, where mostly fine particles exist (Aaltonen et al., 2012).

We thank the reviewer for this comment. Table 5 is added in the revised version of the manuscript, including all the information of the cirrus properties based on ground-based lidar observations reported in the literature for the different latitudes. A latitudinal dependence of the cirrus cloud can be seen. See also answer to question number 22.

The following sentence is added in the revised version of the manuscript:

"The three presented datasets are derived from different latitudinal and climatic sites. In this section we firstly examine the latitudinal dependence of the cirrus geometrical and optical properties. The reported values in literature from previous studies based on lidar grounbased dataset and the retrievals of the current one are listed in Table 5 and plotted in Figure 9 for comparison. We can note, that the cirrus geometrical properties and the lidar ratio values may vary greatly depending on the latitude and an decreasing trend of the geometrical boundaries with the rise of the distance from the equator is obvious, also reported by satellite observations (Sassen et al., 2008). Generally, cirrus layers have been observed up to altitudes of 13km above Gual Pahari, whereas they have only been detected to about 1km lower at the subarctic region and this conclusion is in accordance with the Cloudsat observations (Sassen et al., 2008). Based on the satellite information, the derived cirrus cloud thicknesses was found to be larger in the tropics and decreasing toward the poles. Also from the values reported from groundbased studies, a pattern can be concluded: cirrus cloud geometrical properties peaks around the equator and at midlatitudes sites, with generally decreasing amounts as the poles are approached. On the other hand, the optical properties seem to follow a diverse relation, showing greater values moving to the poles. In our study, larger values of COD 355 and smaller LR were observed for Gual Pahari and Elandsfontein. The larger variability of the optical properties at the two subtropical regions, relative to Kuopio, could be related to the larger and variable aerosol load over these regions."
In the study discussed below, the authors around Kalliopi Artemis Voudouri present a cirrus cloud statistics for several sites where a multi-wavelength Raman lidar system of type Polly-XT was deployed.

The lidar data was analyzed with a newly developed cirrus retrieval algorithm. Cloud boundary detection was based on wavelet transformation of a kind of normalized background-corrected raw signals. For the identified cirrus layers cloud optical properties for 355-nm and 532-nm wavelength were derived and a multiple-scattering correction was applied. The derived statistics of geometrical and optical cirrus cloud properties are diverse.

Clear conclusions could not be drawn.

I see a certain strengths in the manuscript, given the following facts:

- Presentation of a newly developed cirrus cloud identification algorithm

- Retrieval of Raman-based cirrus optical properties for 355 AND 532 nm

- Demonstration of the potential of a network of similar Raman lidar systems for application of one single retrieval scheme

Nevertheless, flaws in the description of the data analysis technique and in the discussion of the statistics dominate my impression while I was reading (several times) through the manuscript. I felt uncomfortable reading through the results section without knowing exactly how the statistics were derived. "Cases" are presented, but how is one case defined?

**See our comment on answer number 13.**

To how many cloud profiles was the wavelet transform applied to get the boundaries?

See our comment on answer number 1. Are the presented optical properties based on Raman or Klett?

See our comment on answer number 11.

These are important questions. Without knowledge about these, the value of the study is very limited.

I thus recommend a major revision of the manuscript, including a second review phase in order to put the study on a more solid footing.

We thank the reviewer for his/her remarks that helped us to improve the manuscript. In the revised version the reviewer's comments have been extensively taken into account, by improving the discussion of many sections (i.e., algorithm, comparison among the different stations) and by improving the figures that lacked of an accurate description. Moreover, parts of the paper have been restructured and all the figures have been reprocessed, as in the present form is not easy to follow the comparisons of the cirrus properties between the different sites.

Below we report the changes included in the revised manuscript as a response to the comments of the reviewer.

**1) 1 – Ch.3; Retrieval Scheme/Fig. 1:**

The presentation of a retrieval scheme should always be done in such a way that others are able to reproduce it. The scheme given in Ch. 3 does not allow for that, because important information is missing: o Was there range-averaging applied? If yes, under which conditions (see e.g. Fig. 1 b vs. 1 c) o What happened to 1-hour intervals not filled entirely by cirrus clouds?

**How were irregularities in the cirrus cloud structure within the 1-hour averaging period treated?**

The reviewer is right that is difficult to follow the algorithm steps with Figure2, as the time averaging presented is different. In the revised version of the algorithm however, we reprocessed the data and Figure 3 presents the 1-hour averaged profiles.

To calculate the cirrus boundaries, the code applies the following steps:

i) The wavelet covariance is calculated for every single profile (every 30s).

ii) The profiles that fulfil the criteria for a cirrus detection (Figure 2, schematic flowchart) are hourly averaged.

(iii) A mean value of the cirrus base and top are attributed to the one- hour processing. Nan values (free of cirrus sets) are not computed to the mean boundaries.

We also calculate the differences within an hour between the bases/top calculated for every 30s profile, and these should not exceed the 0.5km. If differences are greater than this value, we exclude the case. With this assumption, we also exclude cases with large variability of cirrus layers.

**2) "the signal is normalized with a maximum value below 1.5km". What does this mean? Was the signal normalized using the maximum value found between the ground and 1.5 km height (or range)?**

The normalization is applied to ensure the applicability of the method (the threshold critiria for cirrus boundaries) to all the lidar systems. Given that lidar signals are uncalibrated and signal levels from one lidar system to another can be rather different, the normalization ensures the applicability of the criteria used by Baars et al., 2008. We normalized the range-corrected signal by its maximum value found below 1500 m. (below 2500 for Elandsfontein), which is usually the maximum value of the range corrected signal within the Boundary Layer, as proposed by Baars (2008), in order to use the same threshold values for the cirrus boundaries.

3) In Eq. (1): What is z? altitude or range? Is this z the same z as the one in Eq. 3?

I fear that range and height are mixed-up somewhat. Introduce separate variables for height and range where applicable/needed. Also: Is altitude above ground or above sea level (asl)? Was this considered in the statistics (given that the station elevation varies from 190 to 1745 m asl)?

Yes, the altitude in all plots corresponds to height above sea level, and this is considered in the statistics.

4) Eq. 2: What is Csig? Raw signal? Counts? What is Cbg? What is the difference between C and P (Eq. 1 vs. Eq. 2)?

C stands for the lidar raw signal, while P is the signal after applying the SNR filter, the background correction, the range correction and the normalized correction. That is the reason for using different symbols.

5) Case study/Fig. 2: The case study spans over about 4 hours, but the standard averaging period to derive the wavelet and particle depolarization ratio was 1 hour, wasn't it? I propose to show a case study that uses the actual time- and range-resolution used in the cirrus retrieval scheme.

The reviewer is right. In the revised version of the paper we revised Figure 3, with the hourly application of the cirrus retrieval scheme to the case study.

6) According the Figure 1, zero and background levels as well as normalization were applied to the rangecorrected signal. Is this true? Shouldn't at least the background and zero values be subtracted from the raw signal?

We firstly applied the threshold for the SNR, we then corrected the signal for the zero and background, we calculated the range corrected signal and finally, we applied the wavelet.

7) The selected base temperature of <-20°C (see Fig. 1) gives risk to the inclusion of layers of supercooled liquid water into the statistics, as ice formation occurs pre-dominantly via the liquid phase at T<-27°C (Westbrook et al., 2011). Was there any threshold put on the temperature at cloud top? I'd believe a good value for this could be -38°C or so in order to assure that at least at cloud top no liquid water was present any more.

The reviewer is right. In the revised version of the manuscript we applied an additional criteria for classification, regarding the top temperature in our data processing. So, in Figure 2 the schematic flowchart has been changed with the new threshold applied to the top temperature and also all figures have been reprocessed, according to this new value.

We also modified the paragraph in the revised version, which now reads: "Finally, cloud retrievals from the algorithm are classified as cirrus clouds when the following four criteria were met: i) the particle linear depolarization value is higher than 0.25 (Chen at al., 2002; Noel et al., 2002), ii) the altitude is higher than 6km and iii) the base temperature is below -27°C (Goldfarb et al., 2001; Westbrook et al., 2011) and iv) the top temperature is below -38°C (Campbell et al., 2015)."

Campbell, J. R., Vaughan, M. A., Oo, M., Holz, R. E., Lewis, J. R., and Welton, E. J.: Distinguishing cirrus cloud presence in autonomous lidar measurements, Atmos. Meas. Tech., 8, 435–449, doi:10.5194/amt-8-435-2015, 2015.

8) In Fig. 1: Again, what is altitude? Above sea level? Yes, the altitude corresponds to altitude above sea level.

9) In Fig. 1: Particle linear depolarization ratio is used as criteria for cirrus classification. But this parameter requires the detection of particle backscatter coefficient first.

Shouldn't Fig. 1 thus contain an additional column (between CWT and cirrus criteria) that describes the calculation of the optical properties and multiple-scattering correction?

Yes, the reviewer is right. We revised Figure 3, with the hourly application of the cirrus retrieval scheme to the case study and we also added the hourly backscatter profile.

10) In Fig. 2: Why do the cloud boundaries differ between (b) and (c)? Was there vertical smoothing applied to (c)?

Yes, they differ due to the different time averaging and also to the smoothing applied to the optical properties. In the revised version of the manuscript, we changed smoothing to more strict ones and we reprocessed the figure with the hourly application of the wavelet and the hourly retrievals.

11) - Cirrus optical properties:

- During daytime, Klett-Fernald was applied, and during nighttime Raman was applied? Which values went into the statistics of lidar ratio, optical depth and particle depolarization ratio? Both? Only nighttime?

Yes, the reviewer is right. Both values from the two methods are presented in the statistics presented. However, in the revised version, Table 2 and Table 3 have been added, giving the information of the different geometrical and optical values derived from the two methods. See also comment on answer 13.

12) How were reference height and values determined/set?

The determination of the reference height range in the PollyXT software, is made as follows (Baars et al., 2016):

- the user determines the reference height range (zref) from the quicklook of the range corrected signal and provides the sounding file.

- the code calculates the Rayleigh fits (Freudenthaler, 2009) for several zref
- assesses the determined zref
- finds the optimum zref

A similar method is applied in the Single Calculus Chain algorithm for the backscatter calibration (Mattis et al., 2016). In this method, it is also assumed that the height range provided by the user, where the signal or signal ratio has its minimum is closest to the assumed particle-free conditions.

Baars, H., Kanitz, T., Engelmann, R., Althausen, D., Heese, B., Komppula, M., Preißler, J., Tesche, M., Ansmann, A., Wandinger, U., Lim, J.-H., Ahn, J. Y., Stachlewska, I. S., Amiridis, V., Marinou, E., Seifert, P., Hofer, J., Skupin, A., Schneider, F., Bohlmann, S., Foth, A., Bley, S., Pfüller, A., Giannakaki, E., Lihavainen, H., Viisanen, Y., Hooda, R. K., Pereira, S. N., Bortoli, D., Wagner, F., Mattis, I., Janicka, L., Markowicz, K. M., Achtert, P., Artaxo, P., Pauliquevis, T., Souza, R. A. F., Sharma, V. P., van Zyl, P. G., Beukes, J. P., Sun, J., Rohwer, E. G., Deng, R., Mamouri, R.-E., and Zamorano, F.: An overview of the first decade of PollyNET: an emerging network of automated Ramanpolarization lidars for continuous aerosol profiling, Atmos. Chem. Phys., 16, 5111–5137, https://doi.org/10.5194/acp-16-5111-2016, 2016.

Freudenthaler, V.: Lidar Rayleigh-fit criteria, in: EARLINET-ASOS 7th Workshop, available at: http://nbn-resolving.de/urn/resolver.pl?urn=nbn:de:bvb:19-epub-12970-6 (last access: 11 February 2015), 2009.

Mattis, I., D'Amico, G., Baars, H., Amodeo, A., Madonna, F., and Iarlori, M.: EARLINET Single Calculus Chain – technical – Part 2: Calculation of optical products, Atmos. Meas. Tech., 9, 3009–3029, https://doi.org/10.5194/amt-9-3009-2016, 2016.

**13) In the results section, there should be a discussion of Klett-vs-Raman-based results.**

Table 2 in the revised version of the manuscript, shows the average cloud base and top altitudes and the average geometrical thickness for each site separating daytime and nighttime measurements. The averaged geometrical properties are found to be nearly identical above all sites, with differences less than 0.2km. Table 3 shows the averaged lidar ratio values, which found to be nearly identical above all except Gual Pahari site where average nighttime LR is 4sr higher than that of daytime.

"Table 2 summarizes the mean geometrical values calculated for each site. Differences between the mean values of the geometrical properties in the daytime and nighttime measurements are less than 200m for all sites."

"Table 3 summarizes the mean optical values discussed above, for the three sites, separating daytime and nighttime observations. Generally, the averaged optical properties values are found to be nearly identical, except one site (New Delhi), where average nighttime optical properties found higher than that of daytime. But since this dataset is limited, it cannot be used as a reference one."

**14) - Ch 4.02 Multiple Scattering correction:**

- The lidar observations provide Ptot, but P1 is required. Eq. 4 thus contains 2 unknowns: P1, and F(z). How could the authors solve this equation?

To calculate the multiple scattering correction, the code applies an iterative method including the following steps:

i) The measured extinction profile of the cirrus layer is provided.

ii) With the provided effective radius profile of the cirrus layer (linear relation of the effective radius with the cirrus temperature derived from radio soundings) and the effective (measured) extinction coefficient  $\alpha$  par (z), the model provides the ratio P (z)/P (1) (z).

iii) From (2) a first value for the correcting factor F(z) can be worked out.

iv) The iterative procedure continues till the calculation of a stable correcting factor F(z) is found.

v) The corrected extinction can be then calculated from equation (5) in the manuscript and hence the value of lidar ratio.

**15) – Ch. 5.01 Cirrus cloud cover detection:**

- How is a case defined? What does it mean if there were 28 cases observed over Kuopio in April (P7, L175)?

A case is defined as an hourly case. The algorithm searches every set and the ones that fulfill the criteria for cirrus detection, are hourly averaged. See also comment on question 1.

16) Table 2: What is N? Are these the number of hourly samples?

Yes, these are numbers of hourly samples.

**17) – Ch. 5.05: The title can be modified to 'cirrus classification at Kuopio' because the section only deals with this site.**

The reviewer is right. In the revised version of the manuscript this paragraph has been changed and the title of the Section 4.0.4 is "Cirrus classification at Kuopio".

**18) - Ch. 5.06, Line 321:**

Could the decrease of particle LDR with increasing temperature be explained by the sporadic presence of supercooled liquid water?

Generally, the decreasing particle LDR with increasing temperature is believed to reflect the gradual change in basic ice crystal shape, from plates to columns (Noel et al., 2002). Weitkamp also reported that the presence of supercooled water droplets in cirrus is uncommon. Maybe a combination of cloud radar and lidar retrievals can give as more information.

Lidar. Range-Resolved Optical Remote Sensing of the Atmosphere, in the Springer Series in Optical Sciences, edited by Claus Weitkamp

**19) - Conclusions:**

- What conclusion can be drawn on the conversion of cirrus optical properties from 532 nm to 355 nm, considering that such conversion factors might be required to make future 355-nm/532-nm spaceborne lidar observations comperable? Can the authors make suggestions on which aspects future studies should look in more detail?

The assumption that the backscatter and the extinction coefficients for sufficiently large cirrus particles are spectrally independent; that is, the ratio of cloud backscatter coefficients and the ratio of cloud extinction coefficients will both equal unity, is well established (Reagan et al, 2002) and used in satellite processing schemes. But, it is also reported that the measured variability of cirrus color ratios is much larger than previously realized and that measured color ratios are higher in the tropics (Vaughan et al., 2010). From this study, mean values of LR and COD values in Figure 3 can indicate that there is not a significant spectral dependence, derived from groundbased dataset and differences are mainly found to the extinction profiles (also reported by Haarig et al. 2016). Reasons for that deviations could be either an increase in the MS effect with decreasing wavelength, or that the cirrus crystal size distribution could cause stronger extinction at 532 than at the shorter wavelength of 355 nm, or to the different saturation inside the cirrus layer. Figure 11 presenting the color ratios values on 5°C intervals of cirrus mid temperature, indicate an almost stable behavior with temperature. Generally, we can conclude that for higher altitudes, lower spectral dependence is noticed, taking also into account the number of measurements performed at each site. For the Kuopio station, mean BAE is found 1.1±0.9, while for the less extensive dataset of New Delhi the mean value is found 1.5±0.8 and for Elandsfontein the mean value is 1.4±1.1. So, maybe a more representative dataset in the tropics, should be used in order to conclude about the spectral dependence in these regions.

Reagan, J. A., X. Wang, and M. T. Osborn, Spaceborne lidar calibration from cirrus and molecular backscatter returns, IEEE Trans. Geosci. Remote Sens., 40, 2285–2290, 2002.

Vaughan, M. A., Liu, Z., McGill, M. J., Hu, Y., and Obland, M. D., On the spectral dependence of backscatter from cirrus clouds: Assessing CALIOP's 1064 nm calibration assumptions using cloud physics lidar measurements, J. Geophys. Res., 115, D14206, doi:10.1029/2009JD013086, 2010.

In addition to the points addressed above, I recommend a thorough peer-review of spelling and grammar by the co-authors in beforehand to the submission of the revised manuscript.

- Minor comments will be addressed in the revised version.

References:

[revised manuscript text omitted]

---

## Author Response (AR2)

We thank the editor and the reviewers for their fruitful comments. Indeed, all comments have been extensively taken into account in the present form of the manuscript. In what follows, answers to comments are reported just below each related comment.

1) Please introduce all used parameters and describe their meaning. For instance, the parameters in Eq. (1) are not introduced. Also, it is not clear where the temperature profiles used to obtain cirrus top temperatures come from. Please also define what mid temperature is.

The editor is right. All parameters are introduced properly in the revised version of the manuscript.

Also, the following sentences have been added in a separate paragraph.

"The calculation of the corresponding molecular backscatter and extinction profiles was made based on temperature and pressure profiles obtained from radio soundings at each site. Radiosonde observations released at Safdarjung Airport (28.58°N, 77.20°E) in New Delhi, India twice a day, radiosondes from Upington International Airport (28.40°S, 21.25°E), in South Africa and radiosondes launched daily at 06 and 18 UTC at the Jyvaskyla Airport, located to the southwest (62.39°N, 25.67°E) of the lidar station at Kuopio, were used. Mid-temperature is also calculated in our study."

"Mid remperature is defined as the mean temperature between the temperature at base and temperature at top altitude for each cirrus layer."

**2)Please introduce acronyms upon first usage and stick to them afterwards. For instance, LR is used in line 55 without previous introduction.**

The editor is right. The sentence has changed to "The overall mean value of cirrus optical depth was calculated  $0.37 \pm 0.18$ , while the mean Lidar Ratio value (LR), corrected with a constant factor for multiple scattering, was  $31 \pm 15$  sr." Moreover, all the acronyms are introduced the first time they appear in the revised version of the manuscript.

**3)*I'd suggest to exclude the measurements in India from the paper. Ten data points are simply insufficient to draw any conclusions. You could still mention the findings, though.**

Indeed the dataset derived in India, can not be representative of a climatological analysis. However, we consider keeping the measurements, as an indicator of a seasonal dataset, giving emphasis on the small sampling. As mentioned in the literature (Table 5) there are also other studies based on a seasonal dataset in the tropics and thus results for this station should be shown, with the indication that their representativity is limited.

"Indeed the sampling might not be statistically representative of the cirrus cloud properties, but some first results can be discussed."

4) You need to clarify what is meant with case and occurrence. Is a case one 1-h interval or does it refer to one cloud? If it's the former, you need to provide some discussion regarding the connection between 1-h intervals and whole clouds so that the reader knows how to interpret the statistical findings. When you use occurrence, please clarify the reference, i.e. number of all clouds, cases, observation hours...?

The following sentence has been added in the methodology part (Section 3.1): "An hourly lidar measurement of a cirrus case is defined in this study as follows. To calculate the cirrus boundaries, the wavelet covariance is calculated for every single profile (every 30s) and then a mean value of the cirrus base and top are attributed to the one-hour processing. We also consider the differences within an hour between the bases/top calculated should not exceed the 0.5km. With this assumption, we exclude cases with large variability of cirrus layers."

Moreover, the following sentence has been added in Section 4.1: "As cirrus detection in our study we refer to only lidar determined cirrus cases, as these are described in paragraph 3.1 and thus can not be representative of the general seasonal patterns of cirrus occurrence for the areas under study".

This is further discussed in 4.1. "Generally, PollyXT measured almost continuously (24/7) under favor weather conditions and the profiles that fulfil the criteria for a cirrus detection are hourly averaged, so the pattern presented (not shown) is only an indication and biased by the presence of low clouds and rain."

The whole paragraph 4.1 has been changed, giving the number of cases per station and the observation hours performed.

5) There is an abundance of redundant text. I'd suggest to check with the senior authors for parts that can be omitted or shortened. Please also check the grammar while screening the text.

The editor is right. Some sentences were omitted in the revised version.

6) the description of the multiple-scattering correction is not clear. You mention the measured single-scattering extinction profile. Isn't the measured profile the one affected by multiple scattering?

The editor is right, the measured profile is the one affected by multiple scattering. The following changes are made in the revised manuscript:

"The measured extinction coefficient  $a_{eff}$  is then related to the actual (single scattering) coefficient a(z) through the parameter F as shown in Eq. (5) (Wandinger, 1998)."

The whole paragraph 3.3, has been rephrased in the revised version.

7) Please provide proper captions to all figures. The captions should enable the reader to understand the content of the figures. All shown properties should be mentioned. Please also check that all axis titles are okay and refer to proper parameters. For instance, please replace backscatter with backscatter coefficient in Figure 3c. Also, make sure that the caption agrees with the figures.

The editor is right. All figures are revised and corrected for proper captions.

8) I don't think Figure 4 is needed. You can add the number of day and night cases to any of the plots that show monthly statistics.

The editor is right. Figure 4 is removed from the revised text.

9) Figure 7 and related text could be omitted.

The editor is right. The Figure 7 and the related text are removed from the revised text.

10) Please don't show the same results in a figure and a table. Either stick with Figure 9 or Table 5.

We think that both Figure 9 and Table 5 should be included in the revised version. Table 5 in addition to what is shown in the Figure 9 includes all the necessary information from the literature (e.g. references) and Figure 9 shows the latitudinal dependence of cirrus base and top height, and is more illustrative for the reader.

11) The entire discussion about the effect of aerosols in Section 4.3 is somewhat dubious if there is no evidence presented that those aerosols are at cirrus level. I'd suggest to be more cautious with this discussion. The statistics to draw any conclusions for the Indian site are simply not good enough as outlined before.

The editor is right. The following paragraph is added:

[revised manuscript text omitted]